# Free long-chain fatty acids trigger early postembryonic development in starved *Caenorhabditis elegans* by suppressing mTORC1

**Meiyu Ruan**[1☯], **Fan Xu**[1☯], **Na Li**[1], **Jing Yu**[2], **Fukang Teng**[1], **Jiawei Tang**[1], **Cheng Huang**[2], **Huanhu Zhu**[1]*

**1** School of Life Science and Technology, ShanghaiTech University, Shanghai, China, **2** School of Pharmacy, Shanghai University of Traditional Chinese Medicine, Shanghai, China

☯ These authors contributed equally to this work.
* zhuhh1@Shanghaitech.edu.cn

**Data Availability Statement:** All relevant data are within the paper and its Supporting Information files. All RNASeq data have been deposited in BioProject database (accession no. ID

## Abstract

Postembryonic development of animals has long been considered an internally predetermined program, while macronutrients were believed to be essential solely for providing biomatters and energy to support this process. However, in this study, by using a nematode *Caenorhabditis elegans* (abbreviated as *C. elegans* hereafter) model, we surprisingly discovered that dietary supplementation of palmitic acid alone, rather than other abundant essential nutrients such as glucose or amino acid mixture, was sufficient to initiate early postembryonic development even under complete macronutrient deprivation. Such a development was evidenced by changes in morphology, cellular markers in multiple tissues, behaviors, and the global transcription pattern and it occurred earlier than the well-known early L1 nutrient checkpoint. Mechanistically, palmitic acid did not function as a biomatter/energy provider, but rather as a ligand to activate the nuclear hormone receptor NHR-49/80, leading to the production of an unknown peroxisome-derived secretive hormone in the intestine. This hormonal signal was received by chemosensory neurons in the head, regulating the insulin-like neuropeptide secretion and its downstream nuclear receptor to orchestrate global development. Additionally, the nutrient-sensing hub mTORC1 played a negative role in this process. In conclusion, our data indicate that free fatty acids act as a primary nutrient signal to launch the early development in *C. elegans*, which suggests that specific nutrients, rather than the internal genetic program, serve as the first impetus for postembryonic development.

## Introduction

The initiation factors of embryonic development, such as fertilization or parthenogenesis, have been extensively studied. However, it is not clear whether there is a counterpart in postembryonic development. Nutrients are undoubtedly important environmental factors in this

PRJNA1147462; https://www.ncbi.nlm.nih.gov/sra/PRJNA1147462).

**Funding:** This work was supported by the National Natural Science Foundation of China (32170837) (HZ), the National Key R&D Program of China, 2019YFA0802804[HZ], 2021YFA08004801[HZ]), Double First-Class Initiative Fund of ShanghaiTech University (SYLDX0162022)(HZ), Science& Technology Commission of the Shanghai Municipality (16PJ1407400)(HZ), the Recruitment Program of Global Experts of China (Youth)(HZ), and ShanghaiTech Startup program (HZ). Funders did not play any role in the study design, data collection and analysis, decision to publish, or preparation of the manuscript.

**Competing interests:** The authors have declared that no competing interests exist.

**Abbreviations:** CI, chemotactic index; IIS, insulin-like signaling; LCFA, long-chain fatty acid; mmBCFA, monomethyl-branched chain fatty acid; NAC, N-acetyl-L-cysteine; ROS, reactive oxygen species; SCFA, short-chain fatty acid.

process. Under favorable nutrient conditions, animals grow at an optimal speed [1]. When experiencing a nutrient deficiency, the early postembryonic developmental process is usually slowed or even arrested (diapause) until the nutrient condition improves [2]. In recent decades, scientists have discovered multiple nutrient-sensing signaling pathways that critically regulate cell/animal growth during postembryonic development. These pathways include the insulin and insulin-like growth factor signaling (IIS), AMPK, and mTOR pathways [3–5]. Nevertheless, it is widely believed that cell and animal growth initiate only when these major nutrient-sensing pathways are jointly activated by multiple macronutrients (such as carbohydrates, amino acids, or lipids) and even some essential micronutrients [6]. In other words, the availability of each essential macronutrient acts as an AND-gate to initiate the development [7]. This AND-Gate model is theoretically logical and should be conserved in vivo because cells or animals that grow without any major essential macronutrient will die quickly. However, some reports have indicated an alternative situation in vivo [8]. For example, all embryos from rats died in 12 days when the pregnant rat was fed carbohydrate-free diets [9]. On the other hand, rats deprived of essential unsaturated fatty acids started to show skin defects in 3 to 4 weeks (which indicated nutrient deficiency), but they still managed to grow, gain weight, and survive for at least 5 to 6 months before dying of kidney failure [10,11]. These findings suggest that in a given specific developmental stage, only part of key macronutrients is used as the signal to indicate the overall nutrient quality and contributes to the developmental decision in animals. Nevertheless, experimentally proving such a hypothesis in vivo by withdrawing specific macronutrient (we called it a deductive strategy [12]) would be less convincing, as it cannot differentiate between their role as a regulator, or simply an essential biomatter/energy supplier in the development. In addition, it was also technically impractical since most animals would quickly die when a certain essential nutrient is missing from their food [8].

The nematode *Caenorhabditis elegans* provides an ideal model organism to study such a question. First, like many other organisms, the optimal development of *C. elegans* requires many nutrients, such as glucose, amino acids, lipids, and several B vitamins [6,13–15]. Second and most importantly, *C. elegans* can enter a reversible developmental arrest called diapause and survive for an extended period of time under nutrient insufficiency or even complete macronutrient deprivation at multiple developmental stages (referred to as starvation hereafter) [16–18]. Therefore, we can study the role of individual nutrients in *C. elegans* postembryonic developmental regulation by simply supplementing each nutrient individually under starvation. Among them, one of the most studied developmental processes is the early L1 diapause, a stage *C. elegans* is hatched from eggs and quickly enters a reversible diapause when food is absent [18–20]. Several previous studies suggested that certain nutrients, such as glucose, ethanol, and amino acid mixture might act as major nutrient signals to sufficiently promote the developmental process under starvation [12,21,22]. However, either multiple nutrients, or a nutrient whose concentration is far above the physiological condition is required to initiate such a development (see Discussion for details). Thus, it is hard to conclude specific nutrients mainly act as a signal to promote postembryonic development.

In this study, we adopted and modified a previously reported L1 development assay to screen for a single nutrient that could initiate early postembryonic development. By supplying individual macronutrients to newly hatched *C. elegans* under starvation, we surprisingly found only free long-chain fatty acid (LCFA) such as palmitic acid, but not carbohydrates, amino acid mixture (AA mix), or their combination, initiated the very early postembryonic development. Interestingly, this palmitic acid-gated developmental checkpoint occurred earlier than the well-known early L1 diapause stage [21,23]. Our further mechanistic studies revealed that free LCFA, acting as a signal instead of a lipid metabolite, orchestrated sophisticated and cross-tissue regulatory programs involving unusual roles of the mTORC1, nuclear hormone

receptors, peroxisomes derived hormones, sensory neurons, and neuronal hormones sequentially to promote postembryonic development.

## Results

### Palmitic acid specifically initiates *C. elegans* postembryonic development under starvation

We first established an improved assay to identify whether a single nutrient could initiate the early postembryonic development of *C. elegans* under fasting. We took advantage of a commonly used method to synchronize newly hatched *C. elegans* at the first larval stage (L1) by suspending them in the M9 buffer, a saline-like medium without any organic nutrients such as ethanol [22,24]. Under this condition, almost all animals were strictly arrested at the early L1 stage [22,24]. Then, we supplemented individual nutrients into the M9 and observed the animal development after 48 h (Fig 1A). Interestingly, we found that only palmitic acid, but not glucose or amino acid mixture (AA mix hereafter), made animals significantly grow bigger (Fig 1B and 1C). To further distinguish between a real developmental progression from a simple gain of body size, we checked a developmental marker, the maturation of AWC sensory neurons by observing the *str-2*::*GFP* expression pattern (S1A Fig). In early L1 arrested animals, *str-2*::*GFP* showed a dim and symmetric pattern (AWC$^{off}$), while in developing L1 animals, it became much brighter with an asymmetric, or occasionally symmetrically bright pattern (AWC$^{on}$) (S1A Fig) [24,25]. We found only palmitic acid, but not the glucose, amino acid mix, nor even combining them together, could initiate the AWC maturation (Fig 1D and 1E). Additional dietary supplementation of glucose, AA mix, or them together with palmitic acid also did not make a major difference in palmitic acid-induced AWC maturation (S1B Fig). Moreover, a classical chemotaxis experiment [26,27] also indicated that only animals fed with palmitic acid showed mature and functional AWC neurons capable of responding to the chemical cue butanone (Figs 1F and S1C). In addition, to confirm that the palmitic acid-driven AWC neuronal maturation was the result of a real developmental event instead of a stress response that only affected neuronal tissues, we used another epithelial developmental marker *ajm-1*::*GFP*, which labels the division status of hypodermal cells (S1D and S1E Fig) [28,29]. We found that animals fed palmitic acid, but not glucose or AA mix, showed a significant hypodermal cell division (Fig 1G and 1H). Similar results were observed by using another GFP strain (SCM::GFP, which labels seam cell nuclei [30]) under both fluorescent and DIC fields (S1F Fig).

We further used the Q neuroblasts and their descendants (Q cells lineage), one of the best-studied cell lineages in early larval development to precisely stage the development initiated by palmitic acid. We first confirmed that newly hatched *C. elegans* L1 larvae only have 2 Q blast cells, QL, and QR and their division and migration are halted under starvation (in the M9 buffer) [19,23,31] (Figs 1I, 1J, and S1G). Instead, we found palmitic acid initiated the QL and QR differentiation and migration (Fig 1J and 1K). Most QL and QR cells completed their second division and corresponding migration, determined by the AQR (QR.ap), PQR (QL.ap) cells, other Q cell lineages, and their relative position (QR to the anterior and QL to the posterior region) (S1G Fig) [32,33]. However, palmitic acid supplementation could neither promote the third-round division of Q cells, which generated 2 touch neurons AVM and QVM (S1G and S1H Fig), nor initiate the M cells division (S1I Fig) [23,28,34]. These results indicated that the palmitic acid-initiated early developmental events stopped at about 6 to 7 h after egg laying cultured under the normal NGM/*E. coli* condition (6 to 7 h AEL). Based on the somatic cell markers mentioned above, we believe palmitic acid triggers a postembryonic development at the very early stage of L1 (Fig 1L), and its time window (about 3.5 to 6 h AEL) is earlier than the

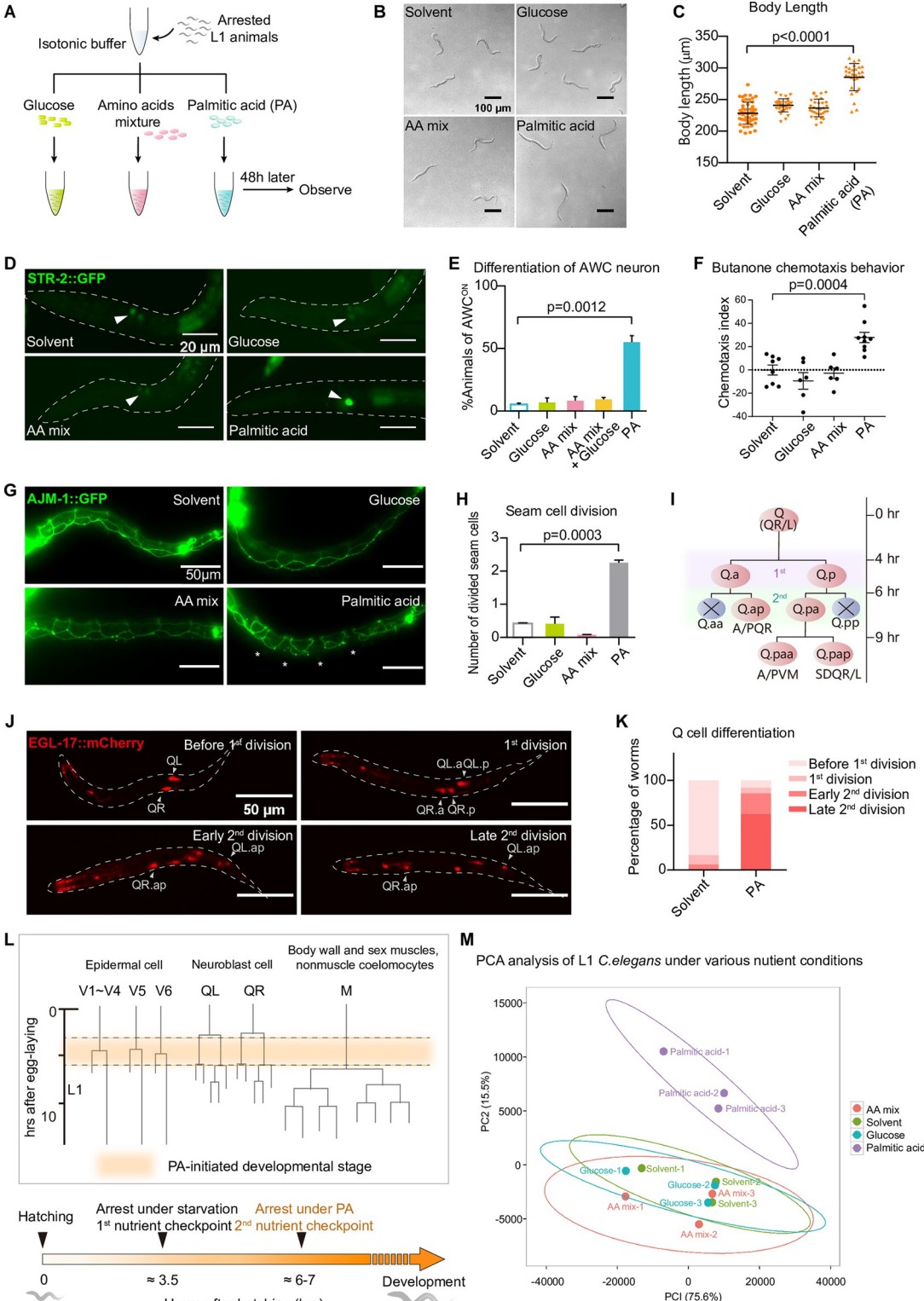

**Fig 1. Palmitic acid promoted the development of arrested L1 *C. elegans* in the isotonic M9 buffer. (A)** A schematic chart of our experimental design for screening key nutrient that initiates postembryonic development in *C. elegans*. Synchronized arrested L1 animals were cultured in M9 buffer with different nutrients and observed after 48–60 h. **(B, C)** Microscopic images (B) and a statistical graph (C) of the body length of the *C. elegans* cultured under various nutrient supplementation. **(D, E)** Fluorescent microscopic images (D) and a statistical graph (E) showing the percentage of animals with mature AWC neurons

cultured under various nutrient supplementation. AWC neurons, marked by STR-2::GFP (*kyIs140*), were indicated by arrowheads. Maturation of AWC neurons was indicated by intense and asymmetric expression STR-2::GFP. **(F)** A dot plot showing the chemotaxis behavior of *C. elegans* in response to butanone, an attractive odor detected by AWC neurons. **(G, H)** Fluorescent microscopic images (G) and a statistical graph (H) showing the development of seam cells, marked by AJM-1::GFP (*jcIs1*). Divided seam cells were indicated by asterisks. **(I–K)** Fluorescent microscopic images (J) and statistical graph (K) showing the development of Q cell lineages, marked by EGL-17::mCherry (*rdvIs1*). Detailed description of Q cell migration was illustrated in cartoon (I) and S1F Fig. For quantification, 48 and 61 animals were counted in the solvent group and palmitic acid (PA) group, respectively. The statistical values from the top to the bottom are 83.33%, 10.42%, 6.25%, 0.00% in solvent group and 8.20%, 6.56%, 22.95%, 62.30% in PA (palmitic acid) group (K). **(L)** A cartoon picture illustrating cell lineages of V1~V6 seam cell, Q cell and M cell in the early postembryonic developmental stage. The orange color showed the time window for palmitic acid-induced development. **(M)** Transcriptome-based principal component analysis among animals cultured under various nutrient supplementation. The palmitic acid-fed group (purple) was prominently different from the other 3 groups. All statistical data are represented as mean ± SEM. C, E, H, Kruskal–Wallis test. H, Ordinary one-way ANOVA with Dunnett's correction. ns, $P > 0.05$, not significant. The data underlying the graphs shown in the figure can be found in S1 Data.

well-known L1 developmental checkpoint gated by the dietary supplements of ethanol and the amino acid mixture together, or the ethanol in the *daf-16(-)* background (after 6 h AEL) [21].

Finally, our transcriptome analyses also confirmed that the gene expression pattern of palmitic acid-fed animals was largely different from fasting animals, or animals supplemented with glucose or amino acid mixture. Instead, it shared a similarity with well-fed animals (Figs 1M and S6C). Taken together, these data showed that free palmitic acid could single-handed initiate *C. elegans* postembryonic development (we named this event "Free palmitic acid initiated Early postembryonic Development Under Starvation", or FEDUS hereafter) in *C. elegans*.

## mTORC1 negatively mediates palmitic acid-induced postembryonic development under fasting

We next investigated the responding tissues and related signaling pathways involved in FEDUS. First, we found anesthetics (such as the nicotinic acetylcholine receptor agonist levamisole, or glutamate-gated chloride channel inhibitor ivermectin) that block food uptake of *C. elegans* by paralyzing the pharynx pumping, could significantly suppress FEDUS (Figs 2A, S2A, and S2B). These data suggest that the palmitic acid needs to be ingested in the intestine, rather than directly sensed by sensory neurons to promote the postembryonic development. Previously, we have found that under the well-fed or dietary restriction condition, intestinal activation of mTORC1, a central hub of nutrient-sensing machinery [4,35,36], was critical to promoting *C. elegans* L1 development [12,24,37,38] under the well-fed condition. Therefore, we tried to suppress FEDUS by knocking down mTORC1. However, we found mTORC1 knocking down by *raga-1(-)* [the ortholog of mammalian RagA/B] or *daf-15(RNAi)* [the ortholog of mammalian Raptor] did not suppress FEDUS (Fig 2B–2D). These data suggest that the activation of mTORC1 was not required for FEDUS. More surprisingly, we found *raga-1(-)* or *daf-15(RNAi)* was able to initiate the development even without the supplementation of free palmitic acid (Fig 2B–2D). We then measured the mTORC1 activity by the relative intestinal nucleolar size labeled by the FIB-1 antibody (because mammalian phosphorylated-S6K antibodies could not recognize its *C. elegans* ortholog RSKS-1) [12,39,40]. Interestingly, we found the mTORC1 activity was further reduced, rather than increased under the palmitic acid supplementation (Fig 2E and 2F). These data suggest that the inactivation of mTORC1 is sufficient to promote the development under starvation. Such an effect seems mTORC1 specific, because mTORC2 mutant *rict-1(-)* [orthologue of Rictor] did not cause a similar effect (S2C Fig). To further confirm such striking findings, we constitutively activated mTORC1 by ubiquitously overexpressing RAGA-1^DA, a dominant active form (Q63L mutation) of RAGA-1 (Mammalian RagA/B) [37] and found it completely blocked FEDUS (Fig 2G). These data indicate that the inactivation of mTORC1 is necessary and sufficient to mediate FEDUS.

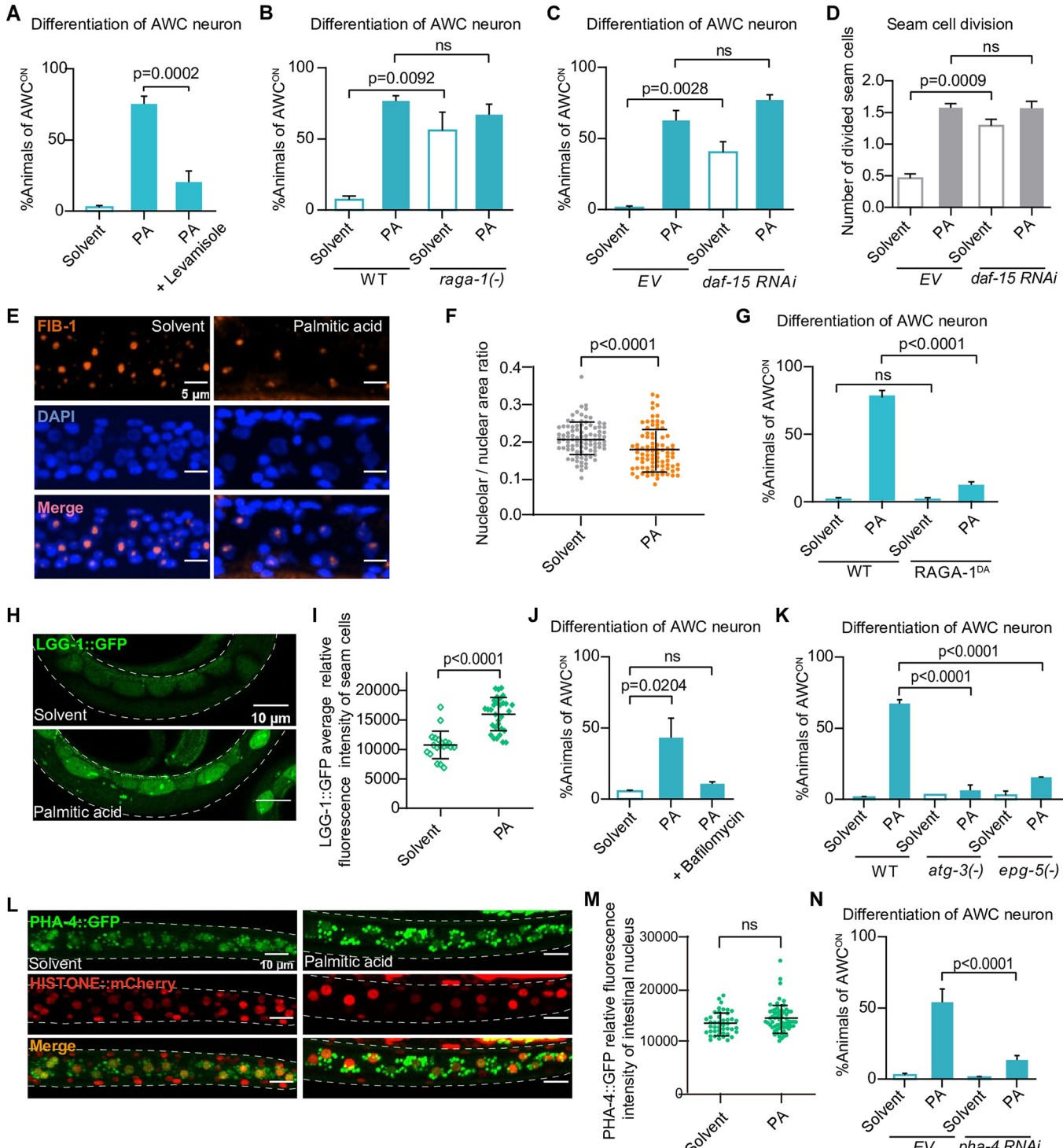

**Fig 2. Inactivation of mTORC1 was necessary and sufficient to mediate FEDUS. (A)** A bar graph showing the anesthetics levamisole (1 mM, added into M9 buffer with worms before FA) significantly suppressed AWC neuron maturation. **(B–D)** Bar graphs showing the *raga-1(ok386)* mutant (B) or RNAi of *daf-15* facilitated AWC neuron maturation (C) and seam cell division (D) in FEDUS. EV, empty vector. **(E, F)** Representative immunofluorescent microscopic images (E) and statistical data (F) showing that the relative nucleolar size was significantly reduced in FEDUS. The nucleoli were labeled with the FIB-1 antibody (orange), and the nuclei were stained with DAPI (blue). **(G)** A bar graph showing a dominant-active mutant of RAGA-1^Q63L (RAGA-1 DA) suppressed AWC neuron maturation. **(H, I)** Fluorescent microscopic pictures (H) and statistical data (I) showing the autophagy level indicated by LGG-1::GFP in epidermal cells. The autophagy elevated in palmitic acid (PA) fed animals (bottom). Dashed lines marked the edge of *C. elegans* body (H). For quantification, the average fluorescence intensity of 3 brightest epidermal seam cells of each animal were counted (I). **(J, K)** Bar graphs showing the percentage of animals with mature

AWC neurons under solvent or palmitic acid (PA) supplementation. Autophagy inhibitor bafilomycin (25 μg/ml, J), *atg-3(bp412)*, or *epg-5(tm3425)* (K) suppressed the AWC neuron maturation in FEDUS. **(L, M)** Fluorescent microscopic images (L) and statistical data (M) showing the intestinal PHA-4::GFP expression. The level of whole body PHA-4::GFP was increased in palmitic acid (PA)-fed animals. The nuclei were labeled with *Prpl-28::histone::mCherry*. **(N)** A bar graph showing the percentage of animals with mature AWC neurons under solvent or palmitic acid (PA) supplementation. RNAi of *pha-4* suppressed AWC neuron maturation in FEDUS. EV, empty vector. Fig F, I, and M are presented as mean ± SD, other statistical data are represented as mean ± SEM. A~D, G, J, K, N, ordinary one-way ANOVA with Tukey's correction. F, I, M, unpaired two-tailed Student's *t* test. ns, $P > 0.05$, not significant. The data underlying the graphs shown in the figure can be found in S1 Data.

## Autophagy is essential but not sufficient for palmitic acid / mTORC1(-)–induced post-embryonic development

We then asked why the inactivation of mTORC1 by free palmitic acid triggered the development under starvation, given a common understanding that mTORC1 activity is relatively low under fasting [12,24,37–39]. We first excluded the less-likely possibility that inactive mTORC1 promoted FEDUS by inhibiting translation, because cycloheximide (a eukaryotic translation inhibitor) supplementation blocked FEDUS, rather than triggered the L1 development under starvation (S2D Fig). Another plausible explanation was that the inactivation of mTORC1 triggered autophagy and generated amino acids and other nutrients from the lysosome-dependent recycling process upon starvation [41]. Interestingly, a recent report suggested that, in mammalian tissue-cultured cells, the mTORC1 activity could be reactivated to a basal level under prolonged starvation to attenuate autophagy, which preserves the capacity for long-term survival, though the cause of such a reactivation was not clear [42]. Therefore, we hypothesized that when the basal mTORC1 activity was inhibited by dietary-supplemented palmitic acid, the autophagy level could be restored and the autophagy-derived amino acids and other nutrients could be used to support the animal development. Indeed, we found the autophagy marker LGG-1::GFP was significantly up-regulated (Fig 2H and 2I) [43]. In addition, blocking autophagy chemically (by bafilomycin), or genetically (by mutation of *atg-3/ATG3 or epg-5/EPG5*), also completely suppressed FEDUS (Figs 2J, 2K, and S2E). The up-regulation of autophagy likely depended on PHA-4 (ortholog of FOXA), a transcription factor that is negatively regulated by mTORC1 [39], because palmitic acid supplementation moderately enhanced the PHA-4::GFP expression [44], and *pha-4(RNAi)* dramatically suppressed FEDUS (Fig 2L–2N). On the other hand, mutation of *aak-2/AMPK*, another autophagy regulator, only moderately affected FEDUS (S2F Fig). Interestingly, chemically enhancing the autophagy by trehalose [45,46] or polyamine [47,48] was not sufficient to activate the development under starvation (S2G and S2H Fig), which was consistent with our finding that AA mix alone could not initiate the development under a similar condition (Fig 1B–1H). These data suggest palmitic acid-induced autophagy is necessary, but not sufficient for FEDUS. Therefore, in addition to its role in generating recycled nutrients such as amino acids as biomaterials by autophagy, palmitic acid may also play a more important role to initiate the development.

## Palmitic acid initiates the postembryonic development independent of its metabolites

We then investigated whether palmitic acid acts as a simple nutrient (to provide biomaterials and energy), or acts as a signal molecule to trigger the development. We first determined the minimal concentration of palmitic acid to initiate FEDUS and found as low as 0.05 mM of palmitic acid was sufficient to initiate the animal development (Fig 3A and 3B). In the contrary, 30 mM glucose, or 60 mM AA mix could not promote a similar development (Fig 3C and 3D). Moreover, dietary supplementation of palmitic acid beta-oxidation metabolites such as acetyl-CoA or its cytosol precursor citrate (which provides energy, or building blocks for the fatty

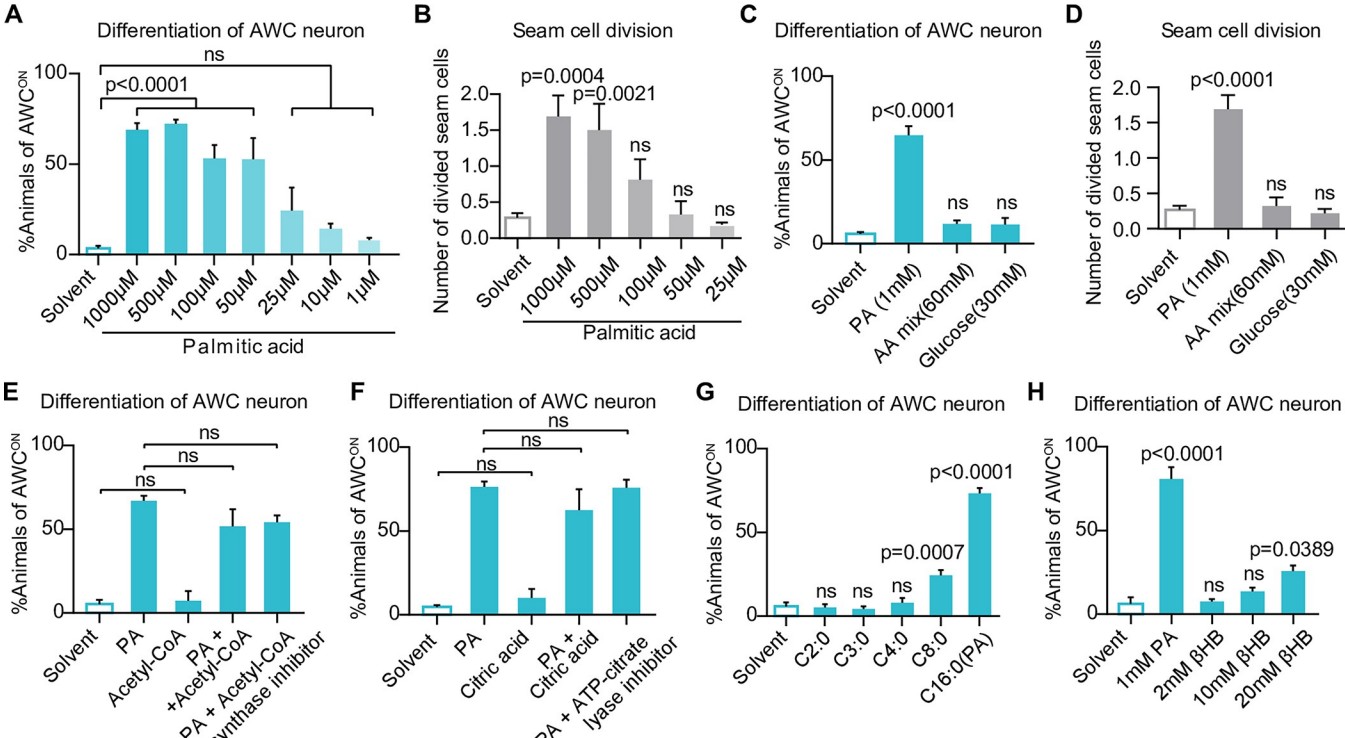

**Fig 3. The development was not attributed to high metabolic energy. (A, B)** Bar graphs showing the percentage of animals with mature AWC neurons (A), or the average number of divided seam cells (B). Gradient concentration of palmitic acid (PA) promoted the maturation of AWC (A) and seam cell division (B). **(C)** A bar graph showing the percentage of animals with mature AWC neurons. Dietary supplementation of 30 mM glucose or 60 mM amino acids mixture could not promote the maturation of AWC neurons. **(D)** A bar graph showing the average number of divided seam cells. Dietary supplementation of 30 mM glucose was not able to promote the division of seam cells. **(E–H)** Bar graphs showing the percentage of animals with mature AWC neurons under various metabolite supplementation. Neither Acetyl-CoA (E), citric acid (F), short-chain fatty acids (G), nor the ketone body beta-hydroxybutyrate (βHB) (H) significantly promoted AWC maturation. Neither Acetyl-CoA synthase inhibitor nor cytosolic Acetyl-CoA ATP citric acid lyase inhibitor (hydroxycitric acid) suppressed FEDUS. All statistical data are represented as mean ± SEM. A, E, F, H, ordinary one-way ANOVA with Tukey's correction. B, C, D, G, ordinary one-way ANOVA with Dunnett's correction. ns, $P > 0.05$, not significant. The data underlying the graphs shown in the figure can be found in S1 Data.

acid/glucose/cholesterol de novo biosynthesis), did not trigger a similar development (Fig 3E and 3F). Blocking cytosolic Acetyl-CoA biogenesis by ATP citric acid lyase inhibitor (hydroxy-citric acid tripotassium hydrate), or by acetyl-CoA synthase inhibitor1 (CAS# 508186-14-9) also could not suppress FEDUS (Fig 3E and 3F). These data indicate palmitic acid neither acts as an energy supplier nor as the acetyl-CoA donor in FEDUS. In addition, we also found short-chain fatty acids (SCFAs), which derive from the palmitic acid beta-oxidation, were not the real development-promoting effector (Fig 3G), even under a much higher concentration (S3A Fig). Similarly, ketone bodies (such as beta-hydroxybutyrate [βHB] or acetoacetate), a class of metabolites biosynthesized from free fatty acids and acting as a major energy source under the dietary restriction condition [49], were also not the major reason of FEDUS (Figs 3H and S3B). These data suggest that palmitic acid does not initiate the development mainly via its common catabolic metabolites under the fasting condition.

Next, we tested whether the palmitic acid initiated the development via its anabolic metabolites. For example, many essential unsaturated fatty acids are derived from palmitic acid by a series of dehydrogenations [50–52]. We found that genetically blocking LCFAs dehydrogenases only showed moderate or no effect on FEDUS (S3C and S3D Fig), suggesting that polyunsaturated fatty acid or their hormone metabolites (such as prostaglandin or endocannabinoids) are not required for FEDUS. Next, we excluded that palmitic acid initiated

the development via the biosynthesis of high-order lipids such as glycerophospholipids via protein lipidation [53,54], because we found genetic mutations of multiple acyl-CoA synthetases (key enzymes for the LCFA to be incorporated into high-order lipid or protein lipidation) did not prominently inhibit the palmitic acid-induced development (S3E Fig). These data suggest FEDUS is not mainly due to the anabolic metabolites of palmitic acid.

## Palmitic acid initiates the postembryonic development via nuclear-hormone-receptor PPARα/NHR-49

The data above implicate palmitic acid itself, but not its metabolites, directly activates the development under fasting. Interestingly, we also found a variety of mono- or polyunsaturated fatty acids triggered the development under starvation (Figs 4A and S3C), suggesting that LCFAs share a common function in the development initiation process. LCFAs have been reported with multiple physiological roles in vitro and in vivo. We first excluded that LCFA acted as an uncoupler of the mitochondrial electron transport chain [55,56] to initiate early development, because other mitochondrial uncouplers such as carbonyl cyanide m-chlorophenyl hydrazone (CCCP) could not induce a similar development like FEDUS (S4A Fig). Moreover, UCP1 inhibitor GTP, or the mutation of its *C. elegans* homolog *ucp-4*, could not inhibit FEDUS (S4B and S4C Fig). In addition, we also excluded LCFAs induced ROS [57] as the main cause of FEDUS, because ROS inhibitor N-acetyl-L-cysteine (NAC) could not affect FEDUS, and the reactive oxygen species (ROS)-induced mitochondria unfolded protein response (UPR$^{mt}$) marker HSP-6::GFP was not changed under FEDUS (S4D and S4E Fig) [58]. As a last resort, we tried 2 less common LCFAs, pentadecanoic acid (C15:0), and heptadecanoic acid (C17:0), individually. We were surprised to find pentadecanoic acid, but not heptadecanoic acid efficiently initiated the postembryonic development (Fig 4B). In addition, we confirmed that dietary supplemented heptadecanoic acid was efficiently absorbed and utilized by *C. elegans* (S4F Fig). Such results immediately caught our attention regarding the well-known nuclear hormone receptor PPARα, since a recent paper reported that free fatty acid pentadecanoic acid, or palmitic acid, but not heptadecanoic acid, act as agonists of mammalian PPARα in vitro and in vivo [59]. Moreover, those unsaturated fatty acids (such as linoleic, linolenic acid, arachidonic, or eicosapentaenoic acid) that initiated FEDUS (Fig 4A) are also known as PPARα agonists [60]. Furthermore, we found that inactive mTORC1 was essential for FEDUS (Fig 2B–2G), which was consistent with a previous finding that mTORC1 inhibition was essential for PPARα activity [61]. Last, we found genetic disruption of NHR-49, the *C. elegans* ortholog of PPARα, could significantly block FEDUS (Fig 4C), and many known genes/pathways that transcriptionally regulated by PPARα/NHR-49 [62,63] were also similarly regulated by palmitic acid supplementation (S6A and S6B Fig). These data suggest PPARα/NHR-49 potentially mediates FEDUS.

## PPARα/NHR-49 mediates FEDUS via the peroxisomal activation

PPARα and its *C. elegans* ortholog NHR-49 are nuclear hormone receptors with pleiotropic functions such as regulating lipid metabolism, aging, and stress response [62,64–68]. NHR-49 usually needs other nuclear hormone receptors (such as NHR-80, NHR-66 or MDT-15) as the coactivator to assert its function [69]. We determined NHR-80 or NGR-66 but not MDT-15 were also essential for FEDUS (Figs 4C, S4G, and S4H). Interestingly, N-acylethanolamines (NAEs), or deficiency of FAAH-1, an enzyme that degrades N-acylethanolamine (another ligand of NHR-49/80 [66]), could not initiate FEDUS (S4I and S4J Fig). These data suggest that the NHR-49/80-dependent FEDUS is specifically induced by free LCFAs.

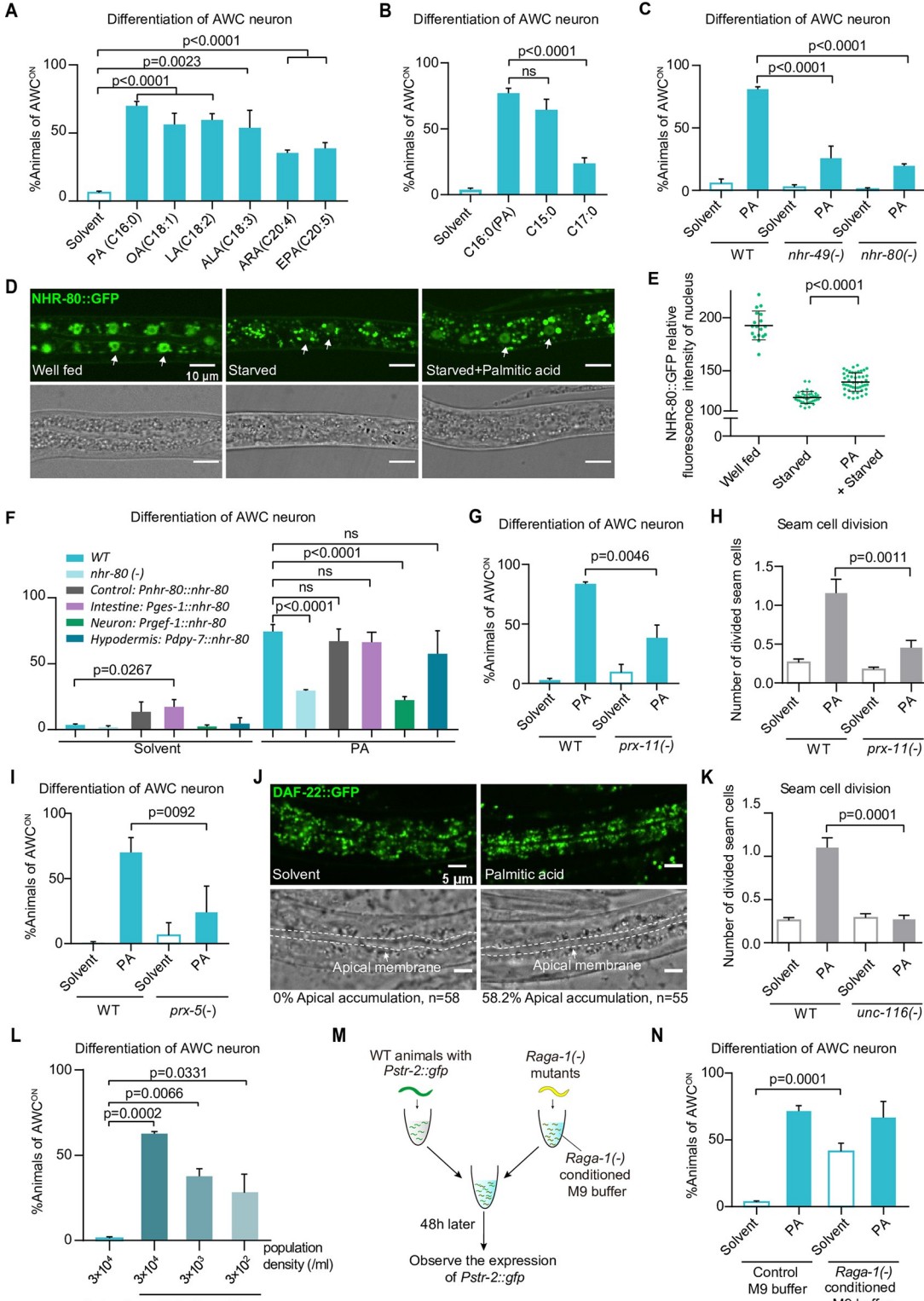

**Fig 4. NHR-49/80 mediated FEDUS via the peroxisomal activation. (A)** Bar graphs showing the percentage of animals with mature AWC neurons. (A) Dietary supplement of unsaturated long-chain fatty acids such as oleic acid (OA, C18:1), linolenic acid (LA, C18:2), α-Linolenic acid (ALA, C18:3), arachidonic acid (ARA, C20:4), or eicosapentaenoic acid (EPA, C20:5) promoted the maturation of AWC neurons. All fatty acids used were 0.3 mM. **(B)** Dietary supplement of pentadecanoic acid (C15:0), but not heptadecanoic acid (C17:0), efficiently promoted the maturation of AWC neurons. **(C)** Loss function mutant of

*nhr-49 (nr2041)* or *nhr-80 (tm1011)* significantly blocked FEDUS. **(D)** Representative fluorescent microscopic images showing the subcellular localization of NHR-80::GFP under various nutrient conditions (upper panel). The nuclei were indicated by arrows. Related bright field microscopic pictures were also shown (bottom panel). **(E)** Statistical data showing that the nuclear relative fluorescence intensity of NHR-80::GFP was significantly down-regulated in starved animals, while palmitic acid (PA) supplementation partially restored its expression. **(F)** A bar graphs showing the percentage of animals with mature AWC neurons. Tissue specific expression of *nhr-80* in the intestine (*Pges-1*) or in the hypodermis (*Pdpy-7*), but not in neurons (*Prgef-1*) rescued the maturation of AWC neurons in *nhr-80 (tm1011)* animals. S is short for solvent. **(G, H)** Bar graphs showing the percentage of animals with mature AWC neurons (G), or the average number of divided seam cells (H). The loss function of *prx-11 (gk959960)* inhibited both of them in FEDUS. **(I)** Bar graphs showing the loss function of *prx-5 (tm4948)* inhibited the AWC neurons maturation. **(J)** Representative fluorescent microscopic images showed the subcellular localization of peroxisomes marked by the *Pdaf-22::gfp::daf-22* knock-in strain. The same samples taken in the bright field were also shown. Dashed lines highlighted the intestinal apical membrane. Palmitic acid supplementation significantly relocated peroxisomes to the intestinal apical region (the percentages of animals with apically localized peroxisomes were listed at the bottom). **(K)** A bar graph showing the average number of divided seam cells. The loss function of *unc-116* inhibited the seam cell division in FEDUS. **(L)** A bar graph showing AWC neurons maturation of animals grew at various population densities. Lower population density significantly decreased palmitic acid-induced AWC maturation. **(M, N)** A cartoon illustration of the experiment design that WT animals grow in M9, or *raga-1(ok386)* mutants-conditioned M9 buffer (details see the Method) (M). A bar graph showing that maturation of AWC neurons was significantly promoted in WT animals grew in *raga-1(ok386)* conditioned M9 (N). Fig E are presented as mean ± SD, other statistical data are represented as mean ± SEM. C, E, F, G, H, I, K, L, N, ordinary one-way ANOVA with Tukey's correction. A, B, ordinary one-way ANOVA with Dunnett's correction. ns, $P > 0.05$, not significant. The data underlying the graphs shown in the figure can be found in S1 Data.

We next elaborated where and how these nuclear hormone receptors mediated FEDUS. We found the nucleic localization of NHR-80 under the fed condition [70] shifted to the cytosol under starvation (Fig 4D). Supplement of palmitic acid significantly increased the nuclear localization and the intensity of NHR-80::GFP (Figs 4D, 4E, and S4K) [70]. We found tissue-specifically restoration of NHR-80 in the intestine (driven by *ges-1* promotor), or in hypodermis (by *dpy-7* promotor), but not in neurons (by *rgef-1* promoter), fully restored FEDUS in *nhr-80(-)* animals (Fig 4F). Furthermore, overexpression of NHR-80 ubiquitously or only in the intestine, was sufficient to partially initiated the AWC maturation in *nhr-80(-)* animals even without palmitic acid supplementation (Fig 4F). On the other hand, there was no difference of the expression/subcellular localization of NHR-49 or NHR-66 between control and palmitic acid-supplementation animals (S4L–S4O Fig), which was in line with a previous report in *C. elegans* that gain-of-function mutants of NHR-49 (constitutively activated without ligands) did not exhibit changes in expression level or subcellular localization [71]. In addition, an NHR-49 gain-of-function allele *nhr-49 (et8)* [71], moderately promoted animal development without palmitic acid supplementation under starvation (S4P Fig). Moreover, knocking down mTORC1 by *daf-15 RNAi* also increased the nuclear localization of NHR-80 to an extent similar to palmitic acid supplementation (S4Q and S4R Fig). Furthermore, by using an HEK293 cell-based reporter assay, we found palmitic acid, but not heptadecanoic acid, could activate the human NHR-49 ortholog PPARα (S4U Fig). Taken together, these data suggest that NHR-49, NHR-66, and NHR-80 are all required for FEDUS, and palmitic acid/inactive mTORC1-induced intestinal nucleus-localization of NHR-80 likely played a more important role in FEDUS. They also implicate that the underlying mechanism could be conserved in humans.

Interestingly, *daf-15 RNAi* could further increase the nuclear localization of NHR-80 in PA-treated animals (S4Q and S4R Fig), and *raga-1(lf)* could moderately increase the nuclear localization of NHR-49 (S4S and S4T Fig). In addition, *raga-1* and PA-supplemented animals shared partial overlap about their impact on NHR-49 downstream targets (S4S and S4T Fig). These differences were likely due to the extent of mTORC1 inactivation by *daf-15* RNAi or *raga-1* was stronger than the LCFA supplementation. Alternatively, fully inactive mTORC1 likely has pleiotropic functions that only part of them was contributed by LCFA supplementation.

## FEDUS is mediated by a secretive hormone/perokine derived from apical intestine-positioned peroxisomes

Next, we investigated the major downstream pathway of NHR-49/80/66 to initiate FEDUS, given several well-known functions of PPARα/NHR-49, such as gluconeogenesis, mitochondrial beta-oxidation of free fatty acid, unsaturated fatty acid biosynthesis, and free fatty acid-induced ROS detoxification [72,73], have been excluded above (S4A–S4E Fig). Moreover, PPARα is known for regulating the mitochondria dynamics [62,69,74]. Though we observed that the mitochondria morphology became more tubular in palmitic acid-fed worms (which further supports that palmitic acids supplementation activated the NHR-49 transcriptional activity), disrupting neither mitochondrial fission by the Dynamin-related protein 1 (Drp1/DRP-1) mutant, nor mitochondrial fusion by the optic atrophy 1 (Opa1/EAT-3) mutant [75] blocked FEDUS (S5A and S5B Fig). These results indicate that mitochondria morphology change is also not the major cause of FEDUS.

Another known function of PPARα/NHR-49 was to mediate peroxisomal proliferation, which is exactly what the name of PPAR comes from [76]. Interestingly, by analyzing our RNA-seq data, we found that peroxisome-related gene/pathways were highly enriched in the palmitic acid-supplemented animals (S6D–S6G Fig). Furthermore, blocking peroxisome function by mutation of *prx-11/PEX11*, which is important for peroxisome proliferation and subcellular relocation [77,78], or by mutation of *prx-5/PEX5* (a key peroxin for the transportation of peroxisomal matrix proteins), also inhibited FEDUS (Fig 4G–4I), even though these mutants were not essential for L1 development [77–79]. These data suggest that NHR-49/80-dependent peroxisome activation plays a critical role in FEDUS.

The critical role of peroxisome in FEDUS is intriguing, especially because we recently have identified that *C. elegans* negatively regulated its L1 development via repositioning of peroxisomes to the apical region of the intestine (e.g., close to the intestinal lumen) by kinesin and facilitating a family of peroxisome-derived secretive metabolite hormones [78]. We then tested whether FEDUS could also be regulated in a similar way. Surprisingly, we found that palmitic acid supplementation relocated peroxisomes to the apical intestine (Fig 4J). Moreover, disruption of peroxisomal apical localization by a mutation of kinesin heavy chain subunit UNC-116 completely disrupted FEDUS (Figs 4K and S5C, which could be rescued by intestinal but neuronal expression of *unc-116* (S5D Fig); suggesting intestinal apical localization of peroxisome is essential for FEDUS). Finally, we found the penetrance of FEDUS was positively regulated by the population density of the animals (Fig 4L), suggesting that a secretive hormone plays an important role in FEDUS. To further verify such a possibility, we grew WT L1 animals with *raga-1(-)* animals (they could initiate their development without palmitic acid, see Fig 2B) together in the isotonic M9 buffer (Fig 4M). We were surprised to find that even without palmitic acid supplementation, about 40% of WT animals started to develop (Fig 4N); which strongly suggests a potential secretive hormone is sufficient to initiate the animal development under starvation. Taken together, these data indicate palmitic acid facilitates the secretion of a development-promoting hormone via relocating peroxisomes to the apical intestine and critically initiate the animal development.

We are curious about the nature of such a peroxisome-derived hormone. PPARα/NHR-49 could activate the biosynthesis of multiple metabolites in peroxisomes, such as metabolites from the sterol de novo pathway, peroxisomal beta-oxidation or alkyl-glycerophospholipids. These metabolites could function as signals to mediate FEDUS [78,80–83]. We first excluded metabolites in the sterol de novo pathway, because (1) supplement of several important intermediate metabolites in this pathway, such as mevalonate or coenzyme Q9 could not initiate the development under fasting (S5E and S5F Fig); (2) inhibition of the sterol de novo pathway

by lovastatin that blocking the rate-limited enzyme HMG-CoA reductase did not significantly affect FEDUS (S5G Fig). We also found peroxisomal beta-oxidation products, such as ascarosides were unlikely to be the signal, since blocking the peroxisomal beta-oxidation by a key Thiolase mutant *daf-22(-)* also did not suppress FEDUS (S5H Fig). Finally, the fact that *fat-1(-)* failed to block FEDUS (S3D Fig) ruled out the possibility that such a hormone was the recently identified small molecule hormone NACQ [84]. Taken together, our data indicate that a type of new peroxisome-derived hormone, or "Perokine," mediates FEDUS.

## The intestinal perokine mediates FEDUS by targeting sensory neurons, downstream insulin-like neuropeptides and the DAF-12 pathway

Next, we studied how the development-promoting perokine further coordinated the whole-body development. *C. elegans* usually senses the hormone/pheromone via their ciliated sensory neurons [78,82,85,86]. We found dysfunction of ciliated neurons by several mutations (*daf-6*, *xbx-1*, or *osm-6*) almost eliminated FEDUS (Fig 5A), suggesting this mystery intestinal peroxisomal signal is also sensed by ciliated sensory neurons. Because many neuropeptides are known to respond to nutrient availability to regulate the L1 development [1], we analyzed whether FEDUS is dependent on the neuropeptide release to control the development. We found a loss-of-function mutant of *unc-31*, an orthologue of human CADPS (calcium-dependent secretion activator) which is responsible for neuropeptide secretion [87], almost completely suppressed FEDUS (Figs 5B, 5C, and S7A). Moreover, *egl-3(-)* (human PCSK2) or *egl-21(-)* (human CPE), 2 enzymes that are essential for neuropeptide processing/production [88,89], also blocked FEDUS (Figs 5D and S7A). These data suggest neuropeptides play a major role in FEDUS. Among many known neuropeptides, insulin-like signaling (IIS) are known to play essential roles in early postembryonic development, especially under fasting [18,90]. Thus, we disrupted IIS by using the IIS receptor *daf-2* (ortholog of human insulin receptor) temperature-sensitive mutant *daf-2(e1370)* and found it completely inhibited the palmitic acid or *raga-1(-)* induced development under fasting (Fig 5E). These data suggest that multiple insulin-like peptides synergistically act at the downstream of palmitic acid and mTORC1 to initiate the development.

The IIS pathway is known to execute its (developmental) function by transducing signal to transcription factor DAF-16 (mammalian FOXO) and nuclear hormone receptor DAF-12 (mammalian NR1I2) [18]. We further tested these 2 pathways and found *daf-12(-)* could initiate the L1 development under complete starvation (Figs 5F and S7B). On the contrary, *daf-16(-)* was not able to initiate FEDUS by itself, a result consistent with the previous finding [21]. However, *daf-16(-)* could promote the development of fasting animals further (beyond AEL 6 to 7 h). A few worms even grew to the L2 larval stage (S7C Fig). Therefore, these data indicate that the IIS-DAF-12 pathway critically mediates FEDUS as the first postembryonic developmental checkpoint, while the DAF-16 pathway may play an important role in the next one, about at the early to middle L1 stage which is sensing amino acid mixture (see Discussion) [21]. Taken together, we propose a model that free LCFAs mainly act as a dominant nutrient signal to initiate *C. elegans* early postembryonic development via a sophisticated gut–brain axis (Fig 5G).

## Discussion

Under starvation, *C. elegans* would enter the protective and reversible diapause [16–18]. Interestingly, several previous works suggested that rather than all essential nutrients together, only 1 or 2 macronutrients could sufficiently activate the postembryonic development in *C. elegans* [12,21,22], suggesting the decision of postembryonic developmental initiation/progression

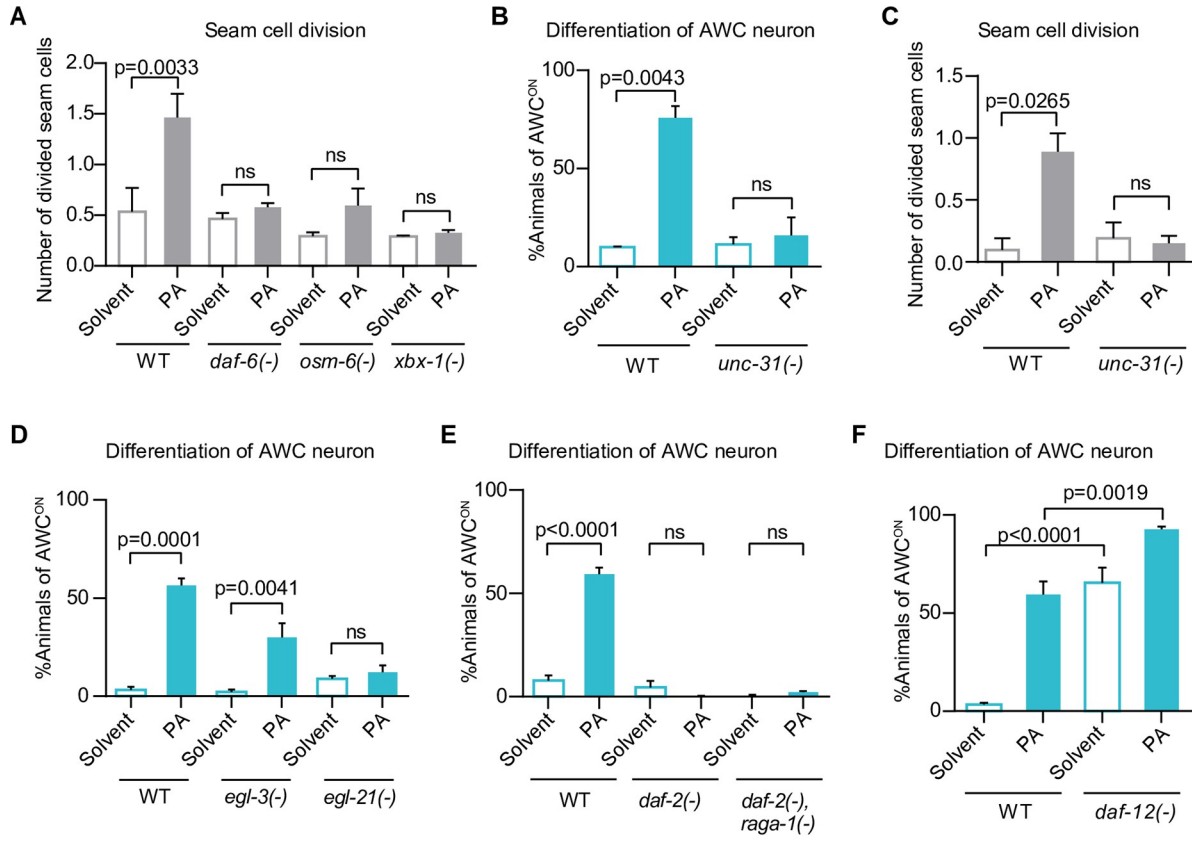

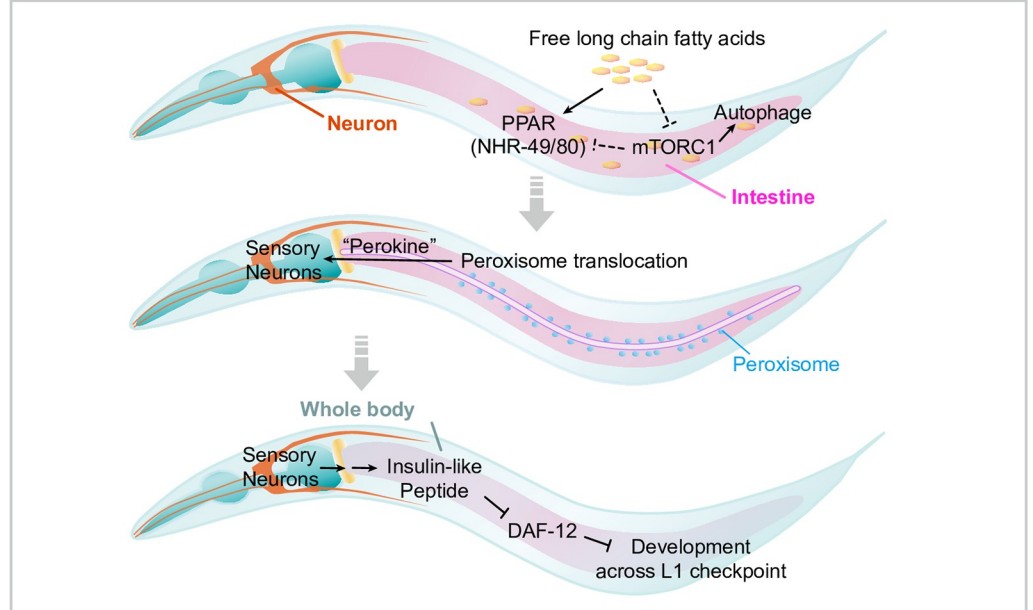

**Fig 5. Ciliated sensory neurons and the insulin-like pathway mediated FEDUS. (A)** A bar graph showing the average number of divided seam cells. The seam cell division was significantly decreased in mutants that block the function of ciliated sensory neurons such as *daf-6 (e1377)*, *xbx-1(ok279)*, or *osm-6(m201)*. **(B, C)** Bar graphs showing the percentage of animals with mature AWC neurons (B), or the average number of divided seam cells (C). The loss function of *unc-31(e928)* inhibited both of them in FEDUS. **(D–F)** Bar graphs showing the percentage of animals with mature AWC neurons. (D) The palmitic acid (PA)-induced maturation of AWC neurons decreased significantly

in neuropeptide processing/production mutants *egl-3(nr2090)* or *egl-21(n476)*. (E) A mutant of insulin-like receptor *daf-2(e1370)* abolished FEDUS of AWC neurons in control or *raga-1(ok386)* animals. (F) Loss function of *daf-12 (rh61rh412)* promoted AWC neuron maturation with/out palmitic acid (PA) supplementation. **(G)** A cartoon model chart showing the cross-tissue mechanism of the free long-chain fatty acid-induced development under starvation (FEDUS). All statistical data are represented as mean ± SEM, ordinary one-way ANOVA with Tukey's correction. ns, $P > 0.05$, not significant. The data underlying the graphs shown in the figure can be found in S1 Data.

could be only made by certain key nutrients. However, following concerns made these studies less conclusive. First, in some of those previous studies, multiple nutrients were needed to reactivate development [21], so one could not exclude the possibility that the development was due to a general improvement of nutritional status. Second, there were some nutrients still in the nutrient-insufficient or dietary-restricted food so one could not exclude the possibility that the supplementation of a specific nutrient molecule promoted the development just by increasing the efficiency of nutrient absorption/utilization [12]. Third, in some works, the quantity of the supplemented nutrient was much higher than its physiological concentration [22], so the result may be subject to alternative explanations. Last, the developmental assay was restricted to a specific type of cells [22], such that the cell division may result from a stress response rather than a physiologically relevant developmental process. Accordingly, to overcome these flaws by an optimized assay, we have discovered an intriguing finding that LCFA such as palmitic acid could initiate *C. elegans* postembryonic development without any other nutrients. The developmental initiation is likely through inactivation of mTORC1, the activation of nuclear hormone receptors NHR-49, its coactivators, and a downstream secretive hormone from the intestinal peroxisome. Such a hormone could be sensed by ciliated sensory neurons and regulate neuronal peptide release and the downstream IIS-DAF-12 pathway to orchestrate the whole animal development. In short, our data indicate LCFA such as palmitic acid, functions as a dominant signal nutrient to initiate the beginning of postembryonic development.

One particular part of this finding is that other nutrient, such as glucose or amino acids, could not achieve a similar effect even under much higher concentrations (30 mM glucose versus 50 μm palmitic acid, a 600-fold difference; 60 mM total AA mix, a 1,200-fold difference) (Fig 3C and 3D). Similarly, high concentrations of other nutrients/metabolites, such as SCFA, citrate, or acetyl-CoA, also could not initiate the development (Fig 3E–3G). It is even against our common sense because theoretically free LCFA could be freely converted from other macronutrients such as glucose, glucogenic/lipogenic amino acids, or citrate endogenously. Therefore, we hypothesize that LCFA has unique advantages as a key signal to initiate early postembryonic development. First, for a living animal that uses its digestive organs to uptake nutrients, LCFA (compared to AA or carbohydrates) may be a more reliable metabolite to indicate the availability of food in nature. After all, LCFA is the major basic building block of the cell membrane and hardly comes from other sources, while amino acids or carbohydrates could derive from the degradation of noncell-based metabolites secreted from other organisms. Second, LCFA is also a basic nutrient directly absorbed by the intestine (the TAG in food has to be hydrolyzed to LCFA (and monoglycerides) by several lipases in the digestive duct before it can be absorbed by enterocytes [91]). On the contrary, most peptides were absorbed in the form of di-/tri-peptide rather than free amino acids [92], while glucose or other monosaccharides are not a major component in food bacteria. Finally, the energy intensity of fatty acid is much higher than amino acid or glucose, which could be critical for very early development. Therefore, animals may develop an LCFA-sensing system in the early developmental stage to adapt to the environmental nutrient status and to regulate the developmental fate accordingly during evolution. Interestingly, newly hatched *C. elegans* seems to develop a mechanism to make sure only dietary LCFA triggers the development by suppressing the

generation of endogenous LCFA (from TAG hydrolysis or de novo fatty acid biosynthesis from amino acids or carbohydrates) under starvation. Our RNAseq data support that genes in TAG hydrolysis or de novo fatty acid biosynthetic pathway are highly up-regulated under palmitic acid supplementation than solvent cohort, amino acid, or glucose supplementation. Of note, a recent paper showed since the mid-embryonic stage, *C. elegans* shifts its major carbon utilization from carbohydrate (glycogen) to neutral lipids to support embryogenesis [104]. Thus, the reason dietary LCFA is used as the sole early postembryonic developmental initiation signal is to satisfy the general metabolic dynamic status (a lipid-prefer mode) inherited from the embryonic stage. Alternatively, the metabolic dynamic shift from carbohydrate to neutral lipids since the mid-embryonic developmental stage could be a secondary evolutional consequence of adapting the postembryonic LCFA-sensing mode. Further molecular mechanistic study is necessary to differentiate those models.

Interestingly, we found several triglyceride lipases were also dramatically up-regulated in palmitic acid-fed animals. These data not only suggest a feed-forward regulation of free LCFA in developmental regulation, but they also imply that the up-regulation of free LCFA level is a physiological process and possibly a rate-limiting step in the early postembryonic development.

Interestingly, there is a significant amount of free LCFA in raw and commercially processed bovine milk (about 1 to 2 mM) and in milk fat (4 mM) [93]. A recent paper also suggests certain omega-6 fatty acids play important roles in activating cardiac metabolism by acting on its nuclear receptor RXR in mice [94], which implies the conservative roles of free LCFA between *C. elegans* and mammals. It would be interesting to further study whether those free LCFAs also play a critical role in the early postembryonic development in mammals, especially via a digestive organ-derived hormone and the gut–brain axis.

Although we found that several nuclear hormone receptors, such as NHR-49 and its coactivators, played essential roles in FEDUS, and that NHR-80 localized to nuclei under palmitic acid supplementation. However, overexpression of NHR-80 or an NHR-49 gain-of-function mutant *nhr-49(et8)* could only moderately initiate the L1 development (in other words, they are not sufficient to effectively initiate the L1 development). We think it was likely because LCFA/inactive mTORC1 also needs to regulate other important targets parallel to NHR-49/80 to initiate FEDUS (such as NHR-66 or other nuclear hormone receptors). In addition, because of technical difficulties in purifying stable NHR-49/80 protein without degradation, our attempts to verify the physical binding between NHR-49/80 and LCFA were unsuccessful. Therefore, we could not exclude the possibility that LCFA indirectly activates the NHR-49/80.

Another interesting finding in this work is that the inactivation of mTORC1 is sufficient to initiate FEDUS. In other words, mTORC1, a common developmental activator, plays a negative role in certain developmental stages. Interestingly, Yu and colleagues have discovered that under prolonged starvation, cells across multiple species showed mTORC1 reactivation, possibly induced by autophagy-derived nutrients [42]. They suggest such a pathway enables a negative feedback regulation of autophagy. However, the signals initiating the autophagy-dependent mTORC1 reactivation and the physiological significance of such regulation remained elusive [42]. Our work implies that the depletion of free LCFA may reactivate mTORC1 and eventually arrest the animal under fasting. Therefore, animals would preserve their nutrient storage and enter the diapause stage, rather than quickly deplete their nutrient storage under starvation. Of note, our previous work found under well-fed conditions, a specific type of LCFA, monomethyl-branched chain fatty acid (mmBCFA) and its endogenous sphingolipid metabolite glucosylceramide, is necessary to activate mTORC1 and support animal development at the L1 stage via Clathrin-dependent vesicle trafficking, intestinal apical polarity maintenance, and apical protein localization [24,37,38,52]. We recently also found

that mmBCFA could activate the mTORC1 to restore *C. elegans* development from the L3 diapause under dietary restriction [12]. We hypothesized that the different roles of mTORC1 could be due to the pleiotropic roles under different nutrient conditions (well-fed/restricted food versus starvation).

Of note, Fukuyama and colleagues found that mTORC1 activity is required to support the developmental progression under both ethanol and amino acid mixture supplementation [21]. In their model, the L1 developmental progression is determined by M cell and P cell division which time point is later than our model (Fig 1L). In addition, they found reactivation of IIS by *daf-16(-)* or reactivation of mTORC1 by *raga-1/ragc-1* could bypass the requirement of amino acid, under the presence of 17 mM ethanol (an early report also showed that *daf-16* mutant could initiate the M cell division of worms starved in S basal, which contains about 22 mM ethanol [90]). Therefore, together with our data, we hypothesize that in early L1 development, the activity of mTORC1 needs to be precisely regulated in a two-step manner. First, mTORC1 needs to be down-regulated to pass the first nutrient checkpoint (guarded by LCFA, or large amount of ethanol). Then, it needs to be reactivated by amino acids and its lipid metabolite mmBCFA/glucosylceramide to pass the second nutrient checkpoint [21,24,37,38,52]. Such a model could also explain that *daf-16(-)* alone cannot initiate the first checkpoint, but when LCFA exists, *daf-16(-)* could help the animals to grow better and advance further (S7C Fig). It would be interesting to study whether free mmBCFA or glucosylceramide is involved in the first developmental progression via the inactivation of mTORC1 under complete starvation.

Finally, it may be worth coming back to the question we raised in the introduction, that is, what is the underlying logic for animals to coordinate their development based on the availability of various nutrients? As we have mentioned above, nutrient deprivation leads to developmental arrest at multiple specific developmental checkpoints, such as early L1, dauer, L3, and L4, which were regulated by multiple signaling pathways such as IIS, mTORC1, and DAF-9 pathways In *C. elegans* [16,19,90,95]. Recent studies further dissected the general concept of nutrients into specific nutrient cues such as amino acids, glucose, mmBCFA, B vitamins, and ethanol, and found they play key roles at these checkpoints [6,21,23,96]. Our current work suggests that only one nutrient, LCFA, is sufficient to initiate the postembryonic development via inactive mTORC1 and DAF-12 pathway. The next L1 checkpoint reported by Fukuyama and colleagues is likely controlled by amino acids through active mTORC1 and IIS pathways [21]. Therefore, it seems that each of those well-known developmental checkpoints may be controlled by a specific nutrient. In other words, animals still need multiple key nutrients to complete their development. However, rather than just following the order of internal genetic signaling to simultaneously provide necessary biomatters/energy in an AND-Gate model, each of those key dietary nutrients could instead dominantly guide relative internal signaling pathways to pass through a specific developmental checkpoint in sequential order. If such a model is proven universally conserved among organisms, we may need a reevaluation of the concept of development and the role of nutrients. After all, the life system originated and evolved from those simple nutrients/metabolites.

## Materials and methods

### *Caenorhabditis elegans* strains and maintenance

*C. elegans* were maintained using standard techniques at 20˚C [97,98] unless otherwise noted. N2 Bristol was used as the wild-type strain. All nematode strains used and generated in this study can be found in S1 Table.

The following transgenic fluorescence strains were used: *kyIs140 [str-2::gfp + lin-15(+)], jcIs1(ajm-1::gfp), rdvIs1[Pegl-17::mCherry], zdIs5 [mec-4::gfp+ lin-15(+)], ayIs7(hlh-8::gfp),*

*zcIs13(Phsp-6::hsp-6::gfp), DA2123[Plgg-1::gfp::lgg-1 + rol-6(su1006)], sydIs031 [Ppha-4::gfp-3\*FLAG], zcIs17[ges-1::gfp(mit)].*

The following mutant strains were crossed with *kyIs140* and/or *jcIs*1 for experiments: *raga-1(ok386), atg-3 (bp412), epg-5(tm3425), nhr49(nr2041), nhr80 (tm1011), prx-11(gk959960)., daf-22 (ok693), daf-6(e1377), osm-6(m201), xbx-1(ok279), unc-31(e928), egl-3(nr2090), egl-21 (n476), daf-2(e1370), daf-12(rh61rh412), daf-16(mgDf50), aak-2(ok524), fat-1(ok2323), fat-2 (wa17), fat-3(ok1126), fat-4(ok958), fat-5(tm420), fat-6(tm331), acs-2(ok2457), acs-5(ok2668), acs-7(tm6781), acs-14(ok3391), acs-17(ok1562), acs-19(tm4853), acs-20(tm3232), acs-22 (tm3236), faah-1 (tm5011), unc-116 (e2310).*

The following transgenic strains were used:

The strain *nhr-80::gfp {lynEx1[(pJG01)nhr-80p::nhr-80::GFP + myo-2p::dsRed]}* was a kind gift from Liang Bin lab. The *Prpl-28/ges-1/rgef-1/dpy-7::raga-1* vectors were previously constructed by Dr. Zhu Mengnan in our lab(21) and injected into *kyIs140* strain.

For the *Pnhr-80::nhr-80* transgenic stain, the genomic DNA, including the full coding region and about 3 kbps of the upstream sequence, was cloned into *pPD95.77* vector. For *Pges-1::nhr-80, Prgef-1::nhr-80*, and *Pdpy-7::nhr-80*, the genomic *nhr80* DNA, including the full coding region, was cloned into *pPD95.77* and driven by the indicated promoters, respectively. For *Pnhr-49::gfp::nhr-49*, the genomic DNA, including the full coding region and 3,637 bps of the upstream sequence, was cloned into *pPD95.77* vector. GFP was inserted at the N-terminal of the DNA.

Plasmids were extracted using QIAGEN kit and injected into indicated *C. elegans* strains. Transgenic *C. elegans* were obtained by microinjecting corresponding plasmids (10 ng/μl) into appropriate strains, and 2 ng/μl *Pmyo-2::gfp* vector was used as the co-injection marker. Animals carrying the marker were considered transgenic strains. Transient transgenic strains carrying the *Pnhr-49::gfp::nhr-49* plasmids were integrated using the UV/TMP method and outcrossed twice.

## Macronutrient preparation

Fatty acids were all dissolved in DMSO and prepared as 100 mM stocks. The solvent shown in most figures was DMSO except otherwise mentioned. Detailed information on fatty acids is provided in S2 Table.

The amino acid mixture and glucose were dissolved in ddH$_2$O. The amino acid mixture was prepared according to the chemically defined *C. elegans* medium (CeMM) [12,14,105]. The detailed recipe of the amino acid mixture is shown in S3 Table.

## Microscopy

Imaging of *str-2::gfp, ajm-1::gfp, hlh-80::gfp, egl-17::mCherry, mec-4::gfp, hsp-6::gfp C. elegans* was performed using an Olympus MVX-ZB10 fluorescence dissecting microscope and a BioHD-C20 CMOS camera. Imaging of the *lgg-1::gfp, nhr-80::gfp, nhr-49::gfp, nhr-66::gfp* animals was conducted using Nikon CSU SORA spinning disk microscope. *Pha-4::gfp* in *C. elegans* was captured using a Zeiss LSM 800 Confocal Laser Scanning Microscopy. The fluorescence intensity was measured using Fiji and analyzed with GraphPad Prism software.

## Developmental assays

To assess AWC cell maturation, well-fed (for at least 3 to 5 generations) *str-2::gfp C. elegans* on NGM plates were collected, washed, and bleached according to standard protocols (each NGM plate should not grow too many animals to avoid food deprivation before bleaching) [97]. Embryos or synchronized L1 *C. elegans* were adjusted to a density of 3 to 6 animals/μl in

M9 buffer (except for the population densities experiment) and then aliquoted equally into experimental groups. Fatty acids were solubilized in DMSO at a concentration of 100 mM for stock solutions. To each 200 μl aliquot of M9 buffer, 3 μl of the 100 mM fatty acid stock solution was added, followed by immediate vortex mixing for 30 to 60 s. Subsequently, 100 μl of M9 (containing synchronized L1 *C. elegans*) were added to the 200 μl fatty acid solution. The final concentration of fatty acid was 1 mM (or other concentration as indicated in this manuscript). Due to the toxicity of high DMSO (used for fatty acid solubilization) concentrations to *C. elegans*, the final DMSO concentration was maintained at less than 1% v/v. After incubating for either 48 h (for synchronous L1 larvae) or 60 h (for bleached eggs), the *str-2::gfp* expression pattern was examined under a fluorescence dissecting microscope.

For the seam cell marker *ajm-1::gfp* and Q cell marker *egl-17::mCherry* assays, L1 animals were prepared as described for *str-2::gfp C. elegans*. The seam cell assay and statistical method were performed as previously described [28]. For each replicate, more than 50 animals were scored. The assay of Q cell migration was performed as previously described (17, 22, 30). Considering the large number of observed animals observed for statistical analysis, we used a fluorescence dissecting microscope instead of a Confocal microscope. Although the detailed pattern of *egl-17::mCherry* positive Q cells lineages could not be clearly visualized under fluorescence dissecting microscope, the general feature of the 4 developmental stages indicated in Figs 1 and S1 could be distinguished clearly based on the number and/or distance of the cells. Of note, fewer than 10% of animals showing abnormal Q lineages and these were not included in the final results.

## Chemotaxis assays

Experiments were performed as previously described with modifications [26,99]; 6 cm plates prepared only with agar and water were used as chemotaxis assay plates. Approximately 100 L1 stage *C. elegans* were placed between the test spot and a control spot on the opposite side of the plate, and 10 μl of 20 mM NaN3 was placed on both spots. After 1 h, all plates were placed in a refrigerator set at 4˚C for subsequent counting. The number of *C. elegans* present at both the test site and control site was recorded, and the chemotactic index (CI) was calculated by subtracting the number of animals at the control site from those at the test site, divided by the total number of animals (S1B Fig). A positive CI value indicates attraction towards the tested substance.

## RNAi experiment

The RNAi-feeding experiments were performed as previously described [100]. dsRNA-expression constructs used were from our previous works [38] or from the ORF-RNAi library (Open Biosystems), and the sequences were validated by Sanger sequencing. In all of the RNAi experiments, L1 *C. elegans* were plated on the RNAi plates. After 3 days, the RNAi-treated adult *C. elegans* were bleached, and the F1 progeny were cultured in M9 buffer and assayed.

## RNAseq

Five animal groups, each with indicated dietary nutrient supplementation, underwent RNA sequencing analysis: solvent (DMSO), palmitic acid, glucose, amino acids mixture (AA mix), and food (*E.coli* OP50). Additionally, mutants of *raga-1(ok386)* were analyzed. Total RNA was extracted using TRIzol reagent (Invitrogen). The sequencing and analysis were performed by Biomarker Technologies Corporation (China).

## Immunofluorescence assay of FIB-1

L1 *C. elegans* were collected and washed 3 times with ddH$_2$O. The immunofluorescence assay of FIB-1 was performed as previously described [12]. The "frozen crack" method was used [101]. Fibrillarin (Abcam ab4566, 1:400) was used as the primary antibody, and goat anti-mouse conjugated to Alexa Fluor 594 (Invitrogen A11005, 1:800) was used as the secondary antibody. DAPI was used for staining nuclei. The fluorescence intensities of intestinal cell nuclear (stained with DAPI) and nucleoli (labeled with FIB-1::GFP) were quantified using Fiji software. Calculations were performed to determine the ratio between the areas of nucleoli and the corresponding cell nuclei.

## Lipid analysis by gas chromatography

About 100 plates (90 mm in diameter) of adult *C. elegans* were collected, bleached, and suspended in M9, supplemented with solvent or heptadecanoic acid (C17:0 fatty acid). After 48 h, the L1 *C. elegans* were washed 3 times with ddH$_2$O. After the final wash, as much water as possible should be removed. The worm pellet was used for fatty acid methylation/extraction, or frozen at −80˚C. The fatty acid methylation/extraction was performed based on a previous report [102]. The method of gas chromatography was described in a previous report [103].

## Culturing WT animals in M9 buffer conditioned with raga-1(ok386) mutants

For the convenience of subsequent observation, we crossed the *raga-1(ok386)* mutant with the *jcIs1(ajm-1::gfp)* strain to generate a *raga-1* mutant line with an epidermal fluorescent marker. We then co-cultured *C. elegans* expressing *str-2::gfp* with the *raga-1(ok386); jcIs1*worms, following the cultivation methods described in the developmental assays. Well-fed adult worms were bleached, and L1 larvae were harvested and co-cultured. The final concentrations of the *str-2::gfp* and *raga-1(ok386); jcIs1* worms were approximately 3 to 6 larvae/µl and 30 to 60 larvae/µl, respectively, in a ratio of approximately 1:10. After a 48-h incubation, the developmental status was evaluated by observing the fluorescence of *str-2::gfp* under a fluorescence microscope.

## Transfection of cultured cells and reporter gene assays

HEK293T cells (CRL-11268, ATCC) were maintained in Dulbecco's Modified Eagle Medium with 10% FBS. For reporter assays, the expression plasmids pCMX-Gal-mPPARα LBD, the Gal4 reporter MH100×4-TK-Luc and pREP7 (Renilla luciferase reporter) were cotransfected using ExFect Transfection Reagent T101 (Vazyme, Nanjing, China). Then, palmitic acid (C16:0), heptadecanoic acid (C17:0), and positive control pirinixic acid (Wy-14643) were added and incubated for 24 h. The luciferase activity was measured using the Firefly Renilla Luciferase Reporter Assay System (MeilunBio, Dalian, China), and the transfection efficiency was normalized according to Renilla luciferase activity.

## Statistical analysis

GraphPad Prism 8.0 was used to perform statistical analysis. Statistical analyses included one-way ANOVA, Kruskal–Wallis test or unpaired Student's *t* test. Data are presented as the means ± SEM unless specifically indicated. At least 3 biological replicates were performed. $P > 0.05$ was considered not significant (ns).

## Supporting information

**S1 Fig. Related to Fig 1. Palmitic acid promoted the development of arrested L1 *C. elegans* under starvation. (A)** Fluorescent microscopic pictures showing the maturation of AWC sensory neurons marked by STR-2::GFP. Mature AWC neurons were indicated by arrowheads. Related to Fig 1D. **(B)** A statistical bar graph showing the percentage of animals with mature AWC neurons under various nutrient supplementations. Mean ± SEM. Ordinary one-way ANOVA. Related to Fig 1E. **(C)** A picture showing the chemotaxis assay to measure whether animals could be attracted to butanone on NGM plates. Related to Fig 1F. **(D, E)** A text description in Knight and colleagues [29] showing the seam cell development of *C. elegans* larvae at different time stages under the fed condition (D). Fluorescent microscopic pictures showing the seam cells (marked by AJM-1::GFP, a marker for adherens junctions) in palmitic acid (PA) fed L1 animals. Divided seam cells were indicated by asterisks (E). Related to Fig 1G. **(F)** Fluorescent microscopic pictures showing the seam cell marker SCM (nuclear signal). The nuclei of seam cells were indicated by arrowheads. Divided seam cells were indicated by asterisks. **(G)** A cartoon picture modified from Ou, G. and Vale, R. D. [33] illustrating the Q cell migration. Related to Fig 1I. **(H)** MEC-4::GFP expressed in 6 gentle touch-sensing neurons including AVM and PVM (third division of Q cell), which could not be observed in either solvent or PA-fed groups. **(I)** Fluorescent microscopic pictures showing M cell division marked by HLH-8::GFP with/out palmitic acid (PA) supplementation. None of M cells was divided in both conditions. The data underlying the graphs shown in the figure can be found in S1 Data. (PDF)

**S2 Fig. Related to Fig 2. Inactivation of mTORC1 was necessary and sufficient for mediating FEDUS. (A, B)** A bar graph showing the percentage of animals with mature AWC neurons (A) and the average number of divided seam cells (B). Animals treated with anesthetics ivermectin (20 ng/ml) exhibited greatly decreased FEDUS. Related to Fig 2A. **(C)** WT or *rict-1(-)* animal, under solvent or palmitic acid (PA) supplementation, showed no difference in the AWC maturation. EV, empty vector. **(D)** A bar graph showing the percentage of animals with mature AWC neurons; 1.5 mM cycloheximide, a eukaryotic translation inhibitor, could suppress FEDUS. PA, palmitic acid. **(E)** The autophagy inhibitor bafilomycin (25 μg/ml) suppressed seam cell division in FEDUS. Related to Fig 2J. **(F)** Loss function of *aak-2 (ok524)* did not suppress FEDUS, as indicated by the percentage of animals with mature AWC neurons. **(G, H)** Autophagy activators, up to100 mM trehalose (G) or 10 mM spermidine (H) treatment did not activate AWC neuron maturation. All statistical data are represented as mean ± SEM. Ordinary one-way ANOVA. ns, not significant. The data underlying the graphs shown in the figure can be found in S1 Data. (PDF)

**S3 Fig. Related to Fig 3. Fatty acid catabolism and fatty-acylation were not required for FEDUS. (A, B)** Bar graphs showing the percentage of animals with mature AWC neurons. Animals supplied with various concentrations of short chain fatty acids were tested (A). Supplementation of acetoacetate, a ketone body, could not initiate the AWC maturation (B). **(C)** A cartoon illustration of the long-chain fatty acid elongation pathway. PA, palmitic acid; POA, palmitoleic acid; STE, stearic acid; cVA, cis-vaccenic acid; OL, oleic acid; LA, linolenic acid; ALA, α-linolenic acid; GLA, γ-linolenic acid; SDA, stearidonic acid; DGLA, dihomo-γ-linolenic acid; ETA, eicosatetraenoic acid; ARA, arachidonic acid; EPA, eicosapentaenoic acid. **(D, E)** Bar graphs showing the percentage of animals with mature AWC neurons. (D) WT and multiple genetic mutants of long-chain fatty acids dehydrogenases [*fat-1(ok2323)*, *fat-2(wa17)*, *fat-3(ok1126)*, *fat-4(ok958)*, *fat-5(tm420)*, *fat-6(tm331)*] in the PUFA biosynthetic pathway

were tested. (E) WT and various mutants of Acyl-CoA synthase [*acs-2(ok2457)*, *acs-5(ok2668)*, *acs-7(tm6781)*, *acs-14(ok3391)*, *acs-17(ok1562)*, *acs-19(tm4853)*, *acs-20(tm3232)*, *acs-22 (tm3236)*] under solvent or palmitic acid (PA) supplement were tested. Among PA-supplement groups, only the *P*-values with significant differences between the groups were labeled. All statistical data are represented as mean ± SEM. Ordinary one-way ANOVA. ns, not significant. The data underlying the graphs shown in the figure can be found in S1 Data. (PDF)

**S4 Fig. Related to Fig 4. NHR-49/80 mediated FEDUS via the peroxisomal activation. (A–D)** Bar graphs showing the percentage of animals with mature AWC neurons. The mitochondrial respiratory chain decoupler CCCP (15 μm) did not initiate the maturation of AWC neurons. Supplementation of 1 mM GTP(B), RNAi of *ucp-4* (C), or the ROS inhibitor NAC (D) could not inhibit the maturation of AWC neurons. EV, empty vector. **(E)** Representative fluorescent microscopic images showing the ROS level indicated by HSP-6::GFP under solvent or palmitic acid supplementation. There was no significant difference between these 2 groups. **(F)** Gas chromatography plot showing the fatty acids profile of L1 animals. Dietary supplementation of heptadecanoic acid (C17:0, red arrows) was indeed absorbed by *C. elegans*. **(G)** Bar graphs showing the loss function of *nhr-66 (ok940)* facilitated AWC neuron maturation in FEDUS. **(H)** Bar graphs showing the RNAi of *mdt-15* did not affect FEDUS. EV, empty vector. **(I)** Supplementation of EPEA (eicosapentaenoyl ethanolamide) or OEA (oleylethanolamide), 2 different NAEs, had no effect on the maturation of AWC neurons. **(J)** Mutation of *faah-1 (tm5011)* had no effect on the maturation of AWC neurons. **(K)** A bar graph showing the percentage of animals with nucleus-localized NHR-80::GFP. Palmitic acid supplementation dramatically increased the nuclear localization of NHR-80::GFP under starvation. Related to Fig 4D. **(L)** Representative fluorescent microscopic images showing the subcellular localization of GFP::NHR-49 under various nutrient conditions (upper panel). Related bright field microscopic pictures were also shown (bottom panel). **(M)** A statistical bar graph showing the percentage of animals with nucleus-localized GFP::NHR-49. For qualification, the fluorescence intensity of 6 intestinal nuclei on one side of each animal were counted (at least 30 animal per condition for each biological replicate). There was no statistical difference between starvation and palmitic acid (PA) supplementation groups. **(N)** Representative fluorescent microscopic images showing the subcellular localization of NHR-66::GFP under solvent (DMSO) and palmitic acid supplementation conditions. **(O)** Statistical data showing no significant difference in the nuclear relative fluorescence intensity of NHR-66::GFP between the 2 groups. **(P)** Bar graphs showing the percentage of animals with mature AWC neurons. *Et7*, *et8*, *et13* were 3 NHR-49 gain-of-function alleles with different substitutive mutations. Nhr-49 (*et8*) moderately promoted animal development without palmitic acid supplementation under starvation. The other alleles showed suppression on FEDUS. **(Q, R)** Fluorescent microscopic pictures (Q) and statistical data (R) showing the subcellular localization of NHR-80::GFP in the presence or absence of *daf-15* RNAi under various nutrient conditions. RNAi of *daf-15* increased the nuclear localization of NHR-80. **(S, T)** Fluorescent microscopic images (S) and statistical data (T) showing the subcellular localization of GFP::NHR-49 in WT or *raga-1(ok386)* mutant worms under various nutrient conditions. **(U)** HEK293 cell-based reporter assay demonstrated that palmitic acid activates the human NHR-49 ortholog PPARα. All statistical data are represented as mean ± SEM. O, unpaired Student's *t* test. Others were ordinary one-way ANOVA. ns, not significant. The data underlying the graphs shown in the figure can be found in S1 Data. (PDF)

**S5 Fig. Related to Fig 4. A secretive hormone derived from apical intestine-positioned peroxisomes mediated FEDUS. (A)** Fluorescent microscopic pictures showing the intestine mitochondria morphology indicated by GES-1::GFP(mit) under treatment of indicated RNAi. Tubular mitochondria morphology was observed in palmitic acid treated worms. RNAi of *eat-3* disrupted the tubular shape and exhibited totally fragment shape. *Drp-1* RNAi worms exhibited disrupted mitochondria structure, tending to form blebs and somewhat tubular shape, compared to the completely starved animals. EV, empty vector. **(B)** Bar graphs showing the percentage of animals with mature AWC neurons. RNAi of *drp-1* and *eat-3* could not inhibit the maturation of AWC neurons. **(C)** A bar graphs showing the loss function of *unc-116 (e2310)* inhibited mature AWC neurons. Related to Fig 4K. **(D)** A bar graph showing tissue specific rescue of UNC-116 in *unc-116 (e2310)* mutants. *Pges-1* and *Prgef-1* were specific promoters in intestine and neuron tissue, respectively. **(E–H)** Bar graphs showing the percentage of animals with mature AWC neurons. Supplementation of 10 mM mevalonic acid (E) or 1 mM coenzyme Q9 (F) did not initiate the maturation of AWC neurons; 1 mM lovastatin (G), an HMG-CoA reductase inhibitor, or (H) a *daf-22* loss-of-function mutant (*ok693*), could not suppress the FEDUS. The data underlying the graphs shown in the figure can be found in S1 Data.
(PDF)

**S6 Fig. Related to Figs 1 and 4. Transcriptional analyses of *C. elegans* under various genetic and nutritional conditions. (A, B)** NHR-49 interaction proteins were analyzed using GeneMANIA (A). The heatmap (B) showing expression of the 20 highest-ranking selected genes under various nutrient conditions. **(C)** A heatmap showing the gene expression of L1 animals under various nutrient conditions. Related to Fig 1M. **(D)** A heatmap showing the peroxisomal β-oxidation related genes of L1 animals under various nutrient conditions. **(E)** A Venn diagram of up-regulated genes (compared to the solvent group) among 3 different expression gene sets (DEGs) shown by 3 colors. A total of 1,124 genes were up-regulated in the solvent (DMSO) vs. palmitic acid (PA) group. **(F)** A KEGG pathway analysis chart of the 1,124 genes depicted in the Venn diagram (E). Peroxisome-related genes are highly enriched. **(G)** Expression changes of genes in the peroxisome group (indicated in red in S5B) were listed. The data underlying the graphs shown in the figure can be found in S1 Data.
(PDF)

**S7 Fig. Related to Fig 5. Ciliated sensory neurons and the insulin-like pathway mediated FEDUS. (A)** A chart showing the neuropeptide processing and secretion pathway. Related to Fig 5B–5D. **(B, C)** Bar graphs showing the average number of divided seam cells. Significant seam cell division was observed in *daf-12* mutant (*rh61rh412*). All statistical data are represented as mean ± SEM. Ordinary one-way ANOVA. ns, not significant. The data underlying the graphs shown in the figure can be found in S1 Data.
(PDF)

**S1 Table. *C. elegans* strains used in this paper.**
(DOCX)

**S2 Table. Fatty acids used in this paper.**
(DOCX)

**S3 Table. Amino acid mixtures.**
(DOCX)

**S1 Data. Numerical data for figures.**
(XLSX)

**S2 Data. RNAseq FPKM data.** Related to S5C Fig.
(XLSX)

## Acknowledgments

We thank Dr. Hong Zhang, Dr. Shohei Mitani, Dr. Gaofeng Fan, Dr. Yingchuan Qi, Dr. Yidong Shen, Dr. Shaobing Zhang, Dr. Hongyun Tang, Dr. Yan Zou, Dr. Shiqing Cai, Dr. Sze-cheng J. Lo, Dr. Min Zhuang and Jianguo Wu, Dr. Lu Zhang, and Dr. Bin Liang laboratories for kindly supplied mutant worm strains, RNAi bacteria strains, technical assistance, or suggestions. We thank *C. elegans* Knockout Consortium and National BioResource Project (NBRP) for mutant worm strains. We thank all of our laboratory members for many helps, especially Dr. Huiyang Xiong and Dr. Mengnan Zhu. We thank Ying Han, Piliang Hao from the Multi-Omics Core Facility (MOCF); Xiaoming Li, Ziwei Yang and Chengyu Fan from the Molecular Imaging Core Facility (MICF); Ying Xiong, Xiaoyue Ren from the Molecular and Cell Biology Core Facility (MCBCF) at the School of Life Science and Technology, Shanghai-Tech University; Lei Zhang from Professional Technical Support Sharing Platform of Core facility (Improvement of Service Capability for Shanghai Proteomics Professional Technical Platform for Severe Diseases) of Shanghai Medical College at Fudan University for providing technical support.

## Author Contributions

**Conceptualization:** Meiyu Ruan, Huanhu Zhu.

**Data curation:** Meiyu Ruan, Fan Xu, Na Li, Jing Yu, Fukang Teng, Jiawei Tang.

**Formal analysis:** Meiyu Ruan, Fan Xu, Na Li, Jing Yu, Fukang Teng, Jiawei Tang, Cheng Huang, Huanhu Zhu.

**Funding acquisition:** Huanhu Zhu.

**Investigation:** Meiyu Ruan, Huanhu Zhu.

**Methodology:** Na Li, Jing Yu, Fukang Teng, Cheng Huang.

**Supervision:** Huanhu Zhu.

**Validation:** Fan Xu.

**Writing – original draft:** Meiyu Ruan, Fan Xu, Huanhu Zhu.

**Writing – review & editing:** Meiyu Ruan, Fan Xu, Jing Yu, Cheng Huang, Huanhu Zhu.

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
