## [Editor Report · Decision Letter 0]

27 Nov 2023

Dear Dr Zhu, 

Thank you for submitting the revision of your manuscript entitled "Free long chain fatty acid solitarily launches early postembryonic development in Caenorhabditis elegans under starvation." for consideration as a Research Article by PLOS Biology.

Your manuscript has now been evaluated by the PLOS Biology editorial staff as well as by the original academic editor and I am writing to let you know that we would like to send your submission out for external peer review.

Once your full submission is complete, your paper will undergo a series of checks in preparation for peer review. After your manuscript has passed the checks it will be sent out for review. To provide the metadata for your submission, please Login to Editorial Manager (https://www.editorialmanager.com/pbiology) within two working days, i.e. by Nov 29 2023 11:59PM.

Kind regards,

Ines

--

Ines Alvarez-Garcia, PhD

Senior Editor

PLOS Biology

---

## [Decision Letter · Decision Letter 1]

13 Feb 2024

Dear Dr Zhu,

Thank you for your patience while your manuscript entitled "Free long chain fatty acid solitarily launches early postembryonic development in Caenorhabditis elegans under starvation" was peer-reviewed at PLOS Biology. Please also accept my apologies again for the time it has taken us to provide you with a decision. Your manuscript has now been evaluated by the PLOS Biology editors, an Academic Editor with relevant expertise, and by two independent reviewers. 

The reviews are attached below. As you will see, the reviewers find the conclusions novel and interesting, but they also raise several issues that would need to be addressed before we can consider the manuscript further for publication. Reviewer 1 asks for several clarifications and further discussion of the conclusions on the role of mTORC1 in promoting development under starvation conditions. In addition, this reviewer thinks that the writing should be improved and that some of the limitations of the data should be acknowledged. Reviewer 2 thinks that you should provide more evidence to confirm that PA acts as a ligand for NHR-49/80 and to show that there is a strong link between the peroxisomal localisation and causal effects. This reviewer also mentions that some of the conclusions are not fully supported and should be toned down, and that the statistical analysis needs further clarification.

In light of the reviews, we would like to invite you to revise the work to thoroughly address the reviewers' reports. Given the extent of revision needed, we cannot make a decision about publication until we have seen the revised manuscript and your response to the reviewers' comments. Your revised manuscript is likely to be sent for further evaluation by all or a subset of the reviewers.

**IMPORTANT - SUBMITTING YOUR REVISION**

3. Resubmission Checklist

a) *PLOS Data Policy*

b) *Published Peer Review*

Sincerely,

Ines

--

Ines Alvarez-Garcia, PhD

Senior Editor

PLOS Biology

Reviewers' comments

Rev. 1:

General comments:

This manuscript describes a series of genetics-based experiments that led to the finding of a highly novel and significant mechanism underlying the regulatory role of a nutrient signal in the initiation of postembryonic development. The novelty of the study lies in both the initial finding of the role of long-chain fatty acid and the mechanism underlying such a role. The ability of one LCFA to initiate postembryonic development was a very surprising and intriguing finding that would impact our thinking about nutrient sensing in general. The finding that mTORC1 negatively regulates the LAFA-triggered initiation of development is incredibly interesting and provocative. The data on other aspects of the mechanism, including roles of two nuclear receptors, peroxisome-produced factors, and sensory neurons, make this quite a complete story. This study reminds us again that nutrient sensing mechanisms need to be explored and analyzed in true in vivo and physiological function-relevant conditions. The authors did a good job in the introduction to explain the rationale of the hypothesis and approach they took in chasing a single nutrient as the growth signal. The experimental data are mostly solid and well represented. With some relatively minor modifications, publication in PLoS Biology is highly recommended.

Specific comments

1. The title of the paper may include information about the role of mTOC1 and nuclear receptors, both of which are among critically significant findings of this work. Adding the information will make the work more attractive to readers.

2. Why would animals use LCFA, not AA or Carb, as the signal to trigger the development at a very early postembryonic stage? Is LCFA a better nutrient signal to represent the overall nutrient and metabolic status? The authors might add some discussion on this.

3. The negative role of TORC1 is truly surprising and interesting to me. The data supporting such a role is excellent. The mention of the positive role of TORC1 in promoting development and growth in a later stage is good, but the discussion may need some modification to make the difference clear to the readers. The positive role was supposed to be there in well-fed animals and deactivated in nutrient deprivation condition, which indicates the role of mTORC1 in promoting postembryonic development. It might be good for readers if the authors would mention the known downstream events in the known positive role described in the previous studies.

4. The authors may need to do more to explain/discuss the downstream roles of mTORC1 in the pathway. I am very glad that autophagy was analyzed, and the authors provide solid data to indicate that autophagy activation was necessary but not sufficient for FEDUS. It seems logical that producing a limited amount of AA and other molecules from autophagy is needed for FEDUS, as cell divisions need at least some of the macronutrients under starvation. Citing a previous study that indicated a link between mTORC1 inhibition and PPARalpha activation in mammals, the authors seem to suggest that the main regulatory role of deactivating mTORC1 was to activate NHR-49/80 in the mechanism for FEDUS. Such a proposed relationship was shown in Figure 5G as a hypothetical interaction (dotted arrow), which was reasonable given the data presented about the roles of the NHR proteins. However, this work did not provide data to support the relationship. Is it technically feasible for the authors to check the NHR activity (nuclear localization, activity of know target genes, etc) under the condition when mTORC1 is artificially deactivated in starved L1 larvae to actually nail the connection? If not, the authors may modify the writing to clearly discuss the two above roles and possibly other roles of mTORC1 in the pathway.

TORC1 is known to activate translation. What would the authors say about translation? Does FIB-1 antibody staining data reflect the status of translation? If mTORC1 activity is still positively correlated with translation under the testing conditions, then the relationship between translation and developmental initiation may be discussed.

5. Figure 4F. The effect of over expression of NHR-80 in the intestine and hypodermis with PA addition was good, but it still just indicates that the NHR is required in at least one of the tissues for the function, and it is not the key evidence that the protein mediates the PA signal. Because the overexpression effect under the condition without PA addition was significant but much weaker, it may not be used as the clear evidence for NHR-80 being the major factor that mediates the PA signal. A good piece of supporting evidence for NHR-80 being a downstream factor of PA was the change in nuclear localization (Fig. 4D-E). The weaker effect of overexpression of NHR-80 in PA(-) condition may be due to the inability of high level expression to constitutively activate the NHR protein. If there is no better way to make a hyperactive NHR-80/49, the authors may want to modify the discussion about the results to point out the limitation/caveat of the data.

6. The authors need to make additional efforts to improve the writing of the manuscript. There are some grammatical errors and awkward/confusing sentences in the manuscript. A few examples are listed below:

- in the writing, the authors should not use "the" in front of FEDUS.

- Last sentence in the abstract: the semi-colon was not used correctly. Should just use period and then start a new sentence.

- page 2, third line from bottom. "These findings suggest that...". The authors may want to consider revising the sentence to be more precise in the message. It is possible all major macronutrients are used as signals in some developmental decision, but in a given specific decision, only one or two were used as the signal to mark the status of overall nutrient levels.

- page 3, line 7, the comma needs to be deleted.

-page 3, line 10, "by simply dietarily supplementing them individually." very awkward. May be changed to: by supplementing each nutrient individually...

- page 4, line 8, at the first larval stage (L1)

page 4, line 4-2 from the bottom, awkward sentence.

page 5, line 2. may change to "We found that animals fed palmitic acid, but not glucose or AA mix, showed ...."

page 14, line 1 under "Discussion". "A few macronutrients" could mean many. There are only a few macronutrients, at most, anyway. The authors need to rewrite the sentence to eliminate the potential confusion. May be just say one or two specific nutrients.

page 15, line 1. "surprisingly intriguing" is awkward, use only one of them. Also need to add a comma after LCFA.

page 16, try not use the word "we guess".

page 16, line 8. The last sentence of the paragraph is confusing. I would delete it. No one would understand without going to detail to explain. Not needed.

- in general, try not use "very" too often. such as "very interesting".

- refence 40 seems to be the same paper of reference 84.

Rev. 2:

The authors describe a seemingly new nutrient checkpoint in C. elegans development that they name FEDUS. The authors report that the addition of palmitic acid alone during starvation can promote development and that this developmental checkpoint occurs earlier than the previously described DAF-16 dependent checkpoint.

They show that the effects of PA can be suppressed (partially) by loss of NHR-49 or NHR-80 and suggest that PA functions as a ligand to activate these transcription factors. They also show that PA treatment promoted relocalization of peroxisomes to the apical side of the intestine and that unc-116 mutation, which inhibits this relocalization, disrupts the FEDUS phenotype. The authors propose that PA promotes the release of a peroxisomal ligand that induces development by acting on chemosensory neurons.

The individual experiments are generally well done and controlled. The description of the FEDUS is interesting and novel. There are some interesting pieces described. However, I find some of the conclusions overstated and the overall model needs more validation to solidify links between individual observations.

Major issues:

1. Altogether, the evidence that PA acts as a ligand for NHR-49/80 is not convincing.

The authors suggest that PA functions as a ligand for NHR-49/NHR-80. The fact that these two NHRs function together has previously been demonstrated. NHR-49 is also reported to function with NHR-66, but there is no mention as to why only NHR-80 was tested. In any case, the authors show suppression of the FEDUS phenotype when either NHR-80 or NHR-49 are absent. However, this suppression is not complete, which does not support the idea that PA functions solely by acting as a ligand for this complex. This is never discussed. In addition, with partial suppression from each mutant, it would seem logical to test the double mutant to validate the idea that they are working as a complex and not independently.

The authors show that NHR-80 is relocalized to the nucleus in response to PA, which does support a potential effect on NHR-80, but it does not show a direct link. PA could just as easily regulate another regulator of NHR-80.

"Taking together, these data suggest that although NHR-49 and NHR-80 are both

required for the FEDUS, only intestinal nucleus-localized NHR-80 critically mediates the FEDUS phenotype" The meaning of "critically mediates" is unclear.

2. The proposed model, that relocation of peroxisomes to the apical membrane facilitates secretion of a peroxisome-derived development-promoting hormone is not well supported. The hypothesis is primarily based on corelative data. How do the authors know that the observed relocalization is a primary response and not secondary to the initiation of growth? For that matter, what is the evidence that the signal for FEDUS comes from the intestine?

The requirement for UNC-116 doesn't provide a great deal of support. UNC-116 functions in many different cell types to aid in the localization of many different structures. If neuronal signaling is needed, how can the authors be sure it isn't functioning there? UNC-116 is also required for the transport of dense core vesicles. Given the requirement for the dense core vesicle binding protein UNC-31 in FEDUS, it seems reasonable to expect that loss of UNC-116 would have the same effect as loss of UNC-31 (loss of either would prevent the release of dense core vesicles).

As such, there is not a strong link between the peroxisomal localization and causal effects.

3. Statistical analysis. In the methods section, it is stated that all experiments, unless noted, were analyzed with a t-test. However, many figures include multiple comparisons, for which a t-test would not be appropriate. Further, there is no mention of a correction for multiple hypothesis testing in any of these figures.

4. The discussion was somewhat unsatisfying. The authors should describe the model in more detail in the discussion. The data with daf-16 are interesting - what does the enhancement with daf-16 mutation mean? Does the addition of PA allow daf-16 mutants to progress through development more rapidly? What does this mean in terms of daf-16 mutants bypassing arrest? How do daf-16 mutants pass the first PA-dependent checkpoint during starvation?

Minor issues

The meanings of statistical indicators (***) are not present in all figure legends; similarly, the statistical test used is often missing from the figure legend.

In the last paragraph of the discussion, the authors discuss the idea of nutritional checkpoints as a novel idea proposed here, but this idea has been proposed by others (and in C. elegans) - these should be referenced.

Methods, strains and transgenes should be properly named (with transgene names and strain names).

Many of the strains used should be referenced with the papers they were originally described in.

Authors note that population density can have an effect on FEDUS, the density of arrested L1s used for experiments should be noted in the methods section

Figure 1F - typo in the panel title

Figure 1M - what are the triangles? Please define in legend.

---

## [Decision Letter · Decision Letter 2]

2 Aug 2024

Dear Dr Zhu,

Thank you for your patience while we considered your revised manuscript entitled "Free long chain fatty acid solitarily launches early postembryonic development in Caenorhabditis elegans by suppressing mTORC1 under starvation" for publication as a Research Article at PLOS Biology. This revised version of your manuscript has been evaluated by the PLOS Biology editors, the Academic Editor and the two original reviewers.

Based on the reviews (attached below), we are likely to accept this manuscript for publication, provided you satisfactorily address the remaining points raised by Reviewer 2 - see my note about the title below. Please also make sure to address the data and other policy-related requests stated below.

In addition, we would like you to consider a suggestion to improve the title:

"Free long-chain fatty acids trigger early postembryonic development in starved C. elegans by suppressing mTORC1"

We expect to receive your revised manuscript within two weeks. 

*Published Peer Review History*

*Press*

Sincerely,

Ines

--

Ines Alvarez-Garcia, PhD

Senior Editor

PLOS Biology

DATA POLICY:

Fig. 1C, E, F, H, K, M; Fig. 2A-D, F, G, I-K, M, N; Fig. 3A-H; Fig. 4A-C, E-I, K, L, N; Fig. A-F; Fig. S1B; Fig. S2A-H; Fig. S3A, B, D, E; Fig. S4A-D, G-K, M, O, P, R, T, U; Fig. S5B-D, F, G and Fig. S6B-H, J, K

CODE POLICY

Reviewers' comments

Rev. 1:

The authors have an impressive job in address all questions from both reviewers. I am very satisfied with their strong effort. The technical aspects of the manuscript has been further improved and the high significance of the work is solidified. The acceptance by PLoS Biology is highly recommended. 

The authors have modified title based on my suggestion on the original manuscript. I think the new title is not good as it may mean that the TOR role was the major/only finding in the paper. In addition, "under starvation" may be at the end for the new title. Here are the two alternative titles I would suggest:

1. Modified new title to: "Free long chain fatty acid solitarily launches early postembryonic development in starved Caenorhabditis elegans and does so by suppressing mTORC1". The only potential weakness is that it may be too long if PLOS Biology has a length restriction. 

2. Go back to the original title, without mentioning the role of TOR. It lacks the mechanistic message as I complained in the original review, but at least it would make it clear regarding the most novel aspect of the finding (FA role). 

Rev. 2:

I thank the authors for the inclusion of additional data that supports some of the findings. I find the manuscript much improved. 

The manuscript text will need a thorough re-write as it still contains many typographical and grammatical errors. In some places, it is also difficult to read (specifically the methods). For example: "Eggs laid naturally by well-fed str-2::gfp C. elegans and L1 C. elegans on NGM plates were collected for growing to adult and subsequent bleaching. Noted that egg/L1 should be adjusted at a suitable number/plate to avoid food deprivation."

Methods should indicate the source of the fatty acids used (I'm not sure if this is in supplemental Table 2 - I was not able to open it). 

The discussion regarding daf-16 is still a little unsatisfying; it seems a bit at odds with findings from the Baugh lab (and others). The authors note in the discussion "Such a model could also explain that daf-16(-) itself cannot initiate the first checkpoint", this is true in the conditions used in this manuscript (in M9 buffer) However, given the FEDUS check point is described as preceding the daf-16 checkpoint, it fails to explain why daf-16 mutants starved in L1 (in S-basal) show development of cells in the M and V lineages (Baugh and Sternberg, Current Biology 2006). A hypothesis is needed to explain how daf-16 mutants in S-basal are bypassing the FEDUS checkpoint. This can be addressed in the text without further experiments.

---

## [Editor Report · Decision Letter 3]

14 Sep 2024

Dear Dr Zhu,

Thank you for the submission of your revised Research Article entitled "Free long chain fatty acids trigger early postembryonic development in starved Caenorhabditis elegans by suppressing mTORC1" for publication in PLOS Biology. On behalf of my colleagues and the Academic Editor, Mark Alkema, I am delighted to let you now that we can in principle accept your manuscript for publication, provided you address any remaining formatting and reporting issues. These will be detailed in an email you should receive within 2-3 business days from our colleagues in the journal operations team; no action is required from you until then. Please note that we will not be able to formally accept your manuscript and schedule it for publication until you have completed any requested changes.

PRESS

Many congratulations and tthanks again for choosing PLOS Biology for publication and supporting Open Access publishing. We look forward to publishing your study. 

Sincerely, 

Ines

--

Ines Alvarez-Garcia, PhD

Senior Editor

PLOS Biology
